# A fast approach for structural and evolutionary analysis based on energetic profile protein comparison

Peyman Choopanian [1], Jaan-Olle Andressoo [1,2,3] ✉ & Mehdi Mirzaie [1,3] ✉

In structural bioinformatics, the efficiency of predicting protein similarity, function, and evolutionary relationships is crucial. Our approach proposed herein leverages protein energy profiles derived from a knowledge-based potential, deviating from traditional methods relying on structural alignment or atomic distances. This method assigns unique energy profiles to individual proteins, facilitating rapid comparative analysis for both structural similarities and evolutionary relationships across various hierarchical levels. Our study demonstrates that energy profiles contain substantial information about protein structure at class, fold, superfamily, and family levels. Notably, these profiles accurately distinguish proteins across species, illustrated by the classification of coronavirus spike glycoproteins and bacteriocin proteins. Introducing a separation measure based on energy profile similarity, our method shows significant correlation with a network-based approach, emphasizing the potential of energy profiles as efficient predictors for drug combinations with faster computational requirements. Our key insight is that the sequence-based energy profile strongly correlates with structure-derived energy, enabling rapid and efficient protein comparisons based solely on sequences.

A thorough understanding of protein function holds paramount importance within the domains of biology, medicine, and pharmacy. While experimental methods exhibit high accuracy in protein function associations, their inherent limitations, such as being time-intensive and expensive, have instigated the exploration of computational alternatives. The evaluation of protein similarity by comparing two proteins has consistently emerged as a key methodology. This assessment plays a pivotal role in uncovering insights into the functions and evolutionary relationships of proteins. Advances in high-throughput technologies have led to the establishment of extensive repositories containing protein sequences, a substantial proportion of which, however, lack annotations[1]. The significant advancements in omics data and the evolution of machine learning techniques have propelled progress in protein research, transitioning from traditional methods like PSI-BLAST[2] to more sophisticated approaches[3]. In the

realm of machine learning research, a crucial step is encoding data as input. Although there is no universal approach, encoding amino acid sequences or structural features has been widely adopted for various protein function predictions, including drug-protein interactions[4], anti-hypertensive peptides[5], and RNA-protein interactions[6]. Despite the diversity in methodologies, the underlying commonality revolves around determining protein similarity either through sequence alignment or structural comparison.

Protein structure is a fundamental feature affecting function, and activity. The energy of a protein structure plays a key role in determining its structure. Knowledge-based potentials, categorized as statistical energy functions, are derived from databases of known protein structures that empirically capture the most probable state of a protein and describe microstates of interactions within protein structure[7–12]. It is generally assuming that a native protein structure is

[1]Department of Pharmacology, Faculty of Medicine, University of Helsinki, Helsinki, Finland. [2]Division of Neurogeriatrics, Department of Neurobiology, Care Sciences and Society (NVS), Karolinska Institutet, Stockholm, Sweden. [3]These authors jointly supervised this work: Jaan-Olle Andressoo, Mehdi Mirzaie. ✉e-mail: jaan-olle.andressoo@helsinki.fi; Mehdi.mirzaie@helsinki.fi

confined to a state with the minimum total energy, and the more similar a structure is to the native state, the closer its total energy is to the native state. However, we take a step beyond this assumption and suggest a hypothesis that two similar proteins possess analogous energy profiles. To evaluate this hypothesis, we assigned an energetic feature vector to each protein, with each entry representing the summation of energies for a specific pair of amino acids. With 20 amino acids in proteins, this resulted in 210 pairwise interaction types. This 210-dimensional vector represents the intricate energy landscape inherent to the structural diversity of proteins. This vector of energies serves as the cornerstone of our analytical approach and provides a robust foundation for further investigative pursuits. Given the current issue of experimentally determining the three-dimensional structures of proteins, estimating energy based on sequence emerges as a crucial consideration. Dostari et al.[13] introduced a method to estimate energy based on amino acid composition. In our study, we drew inspiration from their approach to extract the energy profile based on protein sequence. The concept of using energy profiles to evaluate protein structures was initially introduced by Eisenberg[14], who developed a method for mapping amino acid sequences to structural folds based on energy profiles. This approach enabled an early computational framework for assessing the compatibility of protein sequences with specific structural conformations. Soon after, Wolynes and co-workers[15] expanded this study by applying energy landscape theory, utilizing optimized Hamiltonians to predict protein folding pathways. They introduced the spin-glass model to navigate the complex energy landscape of protein folding, ensuring that the native fold represents a global energy minimum. These pioneering methods laid the groundwork for modern approaches that predict protein structures using energy-based techniques.

The stratification of proteins into distinct folds, superfamilies, and families, guided by evolutionary consanguinity or shared structural and functional attributes, is crucial for precise function prediction. Databases like CATH (Class, Architecture, Topology, Homologous superfamily)[16] and SCOP (Structural Classification Of Proteins)[17] categorize proteins into hierarchical groups based on structural feature, from broad classifications like folds and classes to finer details such as superfamilies and families. To assess profile of energies at various levels, we utilized the ASTRAL40 (95) datasets from SCOPe as a benchmark dataset[18]. Comparing energy and distance between profiles estimated from both sequence and structure revealed a high correlation on protein domains from both ASTRAL40 and ASTRAL95 datasets. UMAP projections provided additional evidence that the profile of energy encapsulates structural information at fold, superfamily, and family levels, as observed through random selections. Our method demonstrated superior performance in terms of both accuracy and computational efficiency compared to currently available tools.

In the realm of structural biology and evolutionary analysis, three-dimensional protein structure classification and the alignment of multiple sequences stand as formidable tools for uncovering structural similarities and deducing phylogenetic relationships. We also evaluated our method to reconstruct evolutionary relationships among proteins from the ferritin-like superfamily that are beyond the "twilight zone"[19] — a sequence similarity range (typically 20−35% identity) that complicates the differentiation between true homologs and random matches due to insufficient sequence conservation. Our findings strongly suggest that a substantial and valuable evolutionary signal is preserved within the profile of energy, serving as a representative indicator of protein structure. To assess the discriminatory capacity of energy profiles in discerning proteins across various species, we chose the spike glycoproteins from three coronavirus species[20]. Our findings indicate that both sequence-level and structural-level energy profiles successfully cluster proteins from distinct species. In a separate analysis, we computed the sequence-based energy profile for a diverse set of bacterial protein families known as bacteriocins. The identification and understanding of these peptides are crucial due to their ecological significance, but their diverse sequences and structures present challenges for conventional identification methods. The BAGEL data set includes 690 proteins[21], each with a length greater than 30 amino acids, providing a comprehensive and challenging benchmark for evaluating peptide identification techniques. Our findings highlight that the energy profile can categorize these proteins based on BAGEL annotation, demonstrating the effectiveness of our method in handling the complexity and diversity inherent in bacteriocin sequences.

The identification of effective drug combinations, essential for treating complex diseases, face challenges due to the combinatorial explosion of potential drug pairs. Cheng et al.[22] introduced a network-based methodology leveraging the human protein-protein interactome to discover clinically effective drug combinations, demonstrating that topological relationships between drug-target modules, as indicated by a separation measure, reflect both biological and pharmacological relationships. In our study, we introduce a separation measure based on the similarity between profile of energies of protein targets, revealing a significant correlation with the separation measure derived from the protein-protein interaction network. This suggests that the profile of energy holds promise as a reliable predictor for drug combinations, requiring only protein sequences and offering quicker computation compared to network-based approaches. Therefore, this study offers a means to characterize and compare proteins using profile of energies, enabling predictions of their structural and functional properties. We further validated the scalability and efficiency of our method by applying it to large datasets, including 4405 coronavirus protein models[23], achieving near-perfect accuracy in sub-family classification and on subsets from the ASTRAL95 dataset, demonstrating superior computational performance.

In this work, we present an approach for characterizing and comparing proteins using profiles of energies derived from structure or sequence data. Our method demonstrates that energy profiles effectively encapsulate structural information, allowing accurate predictions of protein structural and functional properties. We show that the energy profile successfully clusters proteins across diverse taxonomic levels, as evidenced by its performance in classifying spike glycoproteins from coronaviruses and bacteriocins from the BAGEL dataset. Furthermore, our approach predicts evolutionary relationships, even among distantly related proteins in the twilight zone of sequence similarity, and correlates energy-based separation measures with network-derived measures to predict effective drug combinations. Finally, we validate the scalability and efficiency of our method on large datasets, achieving high accuracy and computational performance, thus highlighting its potential for broad applications in protein research.

## Results

Knowledge-based potentials are derived from databases of known protein structures. Various potential functions, such as distance-dependent, dihedral angles, and accessible surface energies leverage information from known protein structures to estimate energies of pairwise interactions[7,8]. In this study, a knowledge-based potential function was developed using a curated dataset of non-redundant protein chains from the Protein Data Bank (PDB), selected for high structural accuracy and diversity as detailed in the Method section. Pairwise distance-dependent potentials were calculated based on atomic interactions, identified through Delaunay tessellation, with energies derived from the frequency of atomic contacts at various distance intervals. The energy between atom pairs was computed following Eq. (3) in the Method section (Fig. 1A). Furthermore, an energy predictor matrix was created to estimate the pairwise energy content solely from amino acid composition (Fig. 1B). Given the 20 amino acids in proteins, Eq. (4) was applied to represent each protein structure using 210 distinct pairwise interaction types (Fig. 1C), leading to the generation of the 210-dimensional Structural Profile of

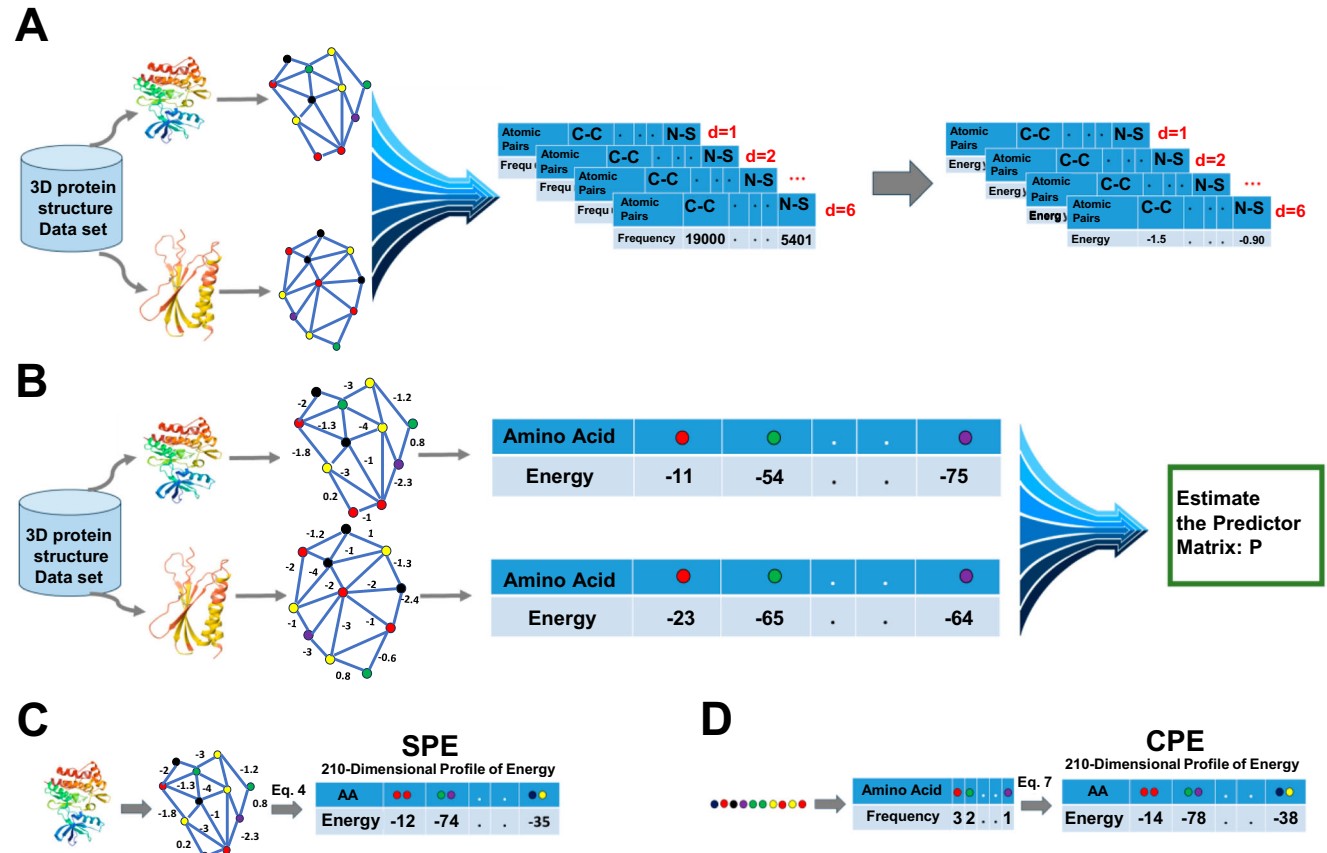

**Fig. 1 | Development of knowledge-based potential function and profile of energy. A** Construction of the knowledge-based potential function. **B** Estimation of the predictor matrix P. **C** Construction of the structural profile of energy (SPE) based on protein structure. **D** Construction of the compositional profile of energy (CPE) based on protein sequence.

Energy (SPE). Additionally, Eq. (7) was employed to compute the Compositional Profile of Energy (CPE) based on protein sequences (Fig. 1D). For each pair of proteins, the Manhattan distance between the profiles of energies is considered a measure of dissimilarity between them.

## Correlation between Energy estimated based on structure and Sequence

To examine the profile of energy at various levels of SCOP, we employed the ASTRAL40 (95) database (version 2.08) from SCOPe as a benchmark dataset, comprising domains with no more than 40% (95%) sequence similarity, as determined by BLAST identity, and filtered for E-value similarity scores[18]. This dataset offers a comprehensive description of structural and evolutionary relationships among proteins from the Protein Data Bank. At first, we calculated energies for protein domains in the ASTRAL40 and ASTRAL95 datasets using both structure- and sequence-based methods. Figure 2A, B depict the strong correlation between total energy derived from structural data (y-axis) and energy estimated from sequence data (x-axis) for the ASTRAL40 and ASTRAL95 datasets. The high correlation coefficient observed suggests that sequence-based energy estimation provides a reliable approximation, which can be effectively applied in cases where the protein structure is unknown. Furthermore, we calculated the total energy for both protein sequences and their corresponding structures using protein domains from the ASTRAL40 dataset and analyzed the differences between these estimates. As shown in Fig. 2C, we specifically examined the correlation between energy differences and protein length. The results indicate no significant correlation, demonstrating

that the accuracy of sequence-based total energy estimates is independent of protein length. This confirms that sequence-based energy calculations can serve as a robust approximation of structural energies across proteins of varying lengths.

For each pair of domains within the ASTRAL40 and ASTRAL95 datasets, the distances between their energy profiles were calculated using both structural and sequence-based energy estimates. In Fig. 2D, E, the x-axis represents the distance between Compositional Profiles of Energies (CPE), while the y-axis represents the distance between Structural Profiles of Energies (SPE). The strong correlation observed between the two approaches indicates that sequence-based energy estimation is sufficiently reliable. To further support this conclusion, we extended our analysis to investigate energy discrepancies across all interaction types. For each interaction type, we calculated the differences between energy estimates derived from sequence and structure. As shown in Fig. 2F, 96% of the interaction types displayed a correlation of less than 0.5 between energy differences and protein length, indicating that protein length does not significantly affect the accuracy of energy estimates for most interaction types. This reinforces our conclusion that sequence-based energy approximations are robust across diverse protein interactions. Supplementary Fig. 1 provides scatter plots for all 210 interaction types. However, while protein length does not appear to influence accuracy, we acknowledge that protein complexity—such as folding patterns, structural heterogeneity, and conformational dynamics—may indeed play a role in the precision of energy estimates. Therefore, exploring the impact of protein complexity on energy estimations will be a valuable direction for future research.

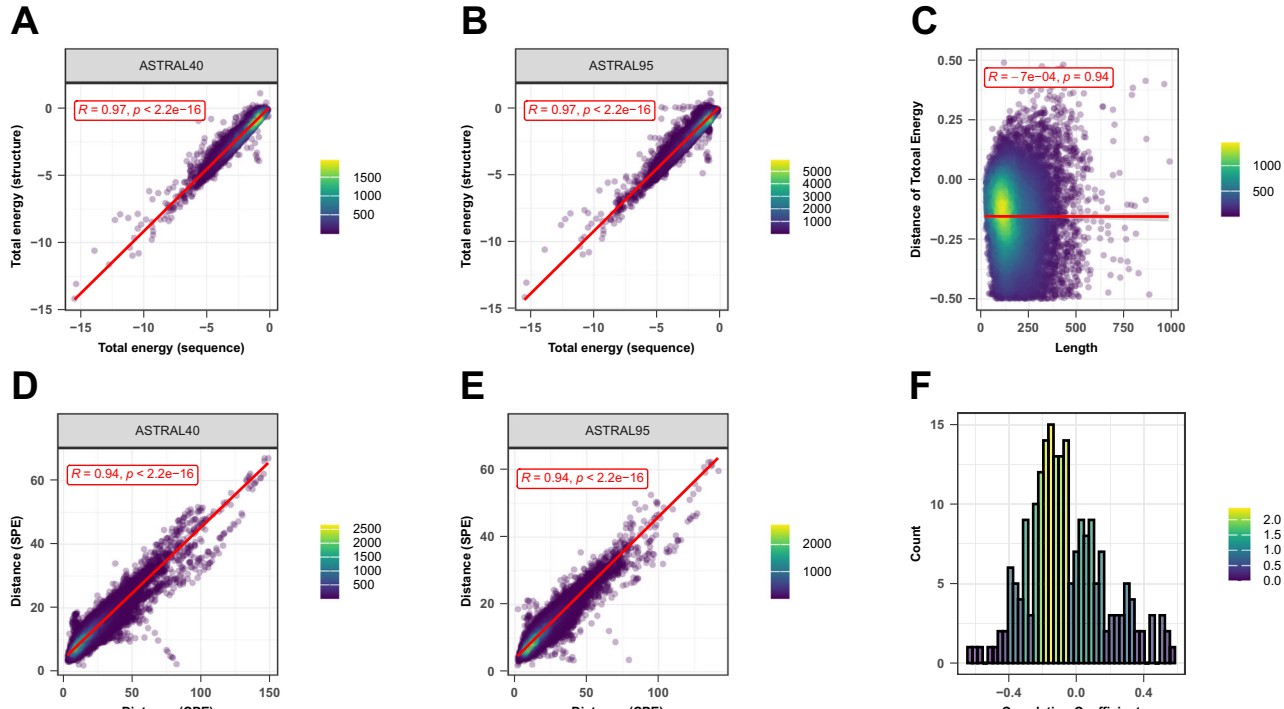

**Fig. 2 | Sequence-Structure relationship.** Two-sided Pearson correlation comparing total energy estimates derived from protein sequence (X-axis) and protein structure (Y-axis) for protein domains in the (**A**) ASTRAL40 data set and (**B**) ASTRAL95 dataset. **C** Two-sided Pearson correlation between the difference in total energy from sequence and structure (Y-axis) and protein length (X-axis). The red lines represent the least squares regression line, and the gray shaded area represents the 95% confidence intervals around the regression line. Each point in (**A**–**C**) represents a protein domain. Two-sided Pearson correlation comparing the distances of profile of energy derived from sequence (X-axis) and structure (Y-axis) for all pairs of domains in D) ASTRAL40 and E) ASTRAL95 datasets, respectively. Each point in (**D**, **E**) represents a pair of protein domain. In plots (**A**–**D**), R indicates the correlation coefficient, and p shows the p-value. The exact p-value is less than 10e-16, which is below the precision threshold of standard statistical computations. **F** Histogram showing the distribution of correlation coefficients between the difference in energy estimates (from sequence and structure) and protein length across all 210 pairwise interactions. Source data are provided as a Source Data file.

The stability, mutational robustness, and design adaptability of α-helices relative to β-strands in natural proteins have been widely acknowledged in scientific literature. To investigate this phenomenon, Supplementary Fig. 2 presents the distribution of total energy within protein domains from the ASTRAL40 and ASTRAL95 datasets, categorized into four structural SCOP classes: all-alpha, all-beta, alpha + beta, and alpha/beta. Total energies, normalized by protein length, are analyzed to discern patterns across these structural classes. The figure highlights significant differences in total energy among domains with different structural compositions, suggesting diverse energetic landscapes associated with distinct protein structures. This observation is consistent with similar trends observed in energy estimations derived from sequence information (Supplementary Fig. 2B).

**Unveiling the energy patterns across SCOP hierarchy**
We visualized energy profiles derived from sequence and structure for domains within the all-alpha and all-beta classes. As shown in Supplementary Fig. 3, UMAP embeddings effectively capture structural characteristics distinguishing all-alpha and all-beta domains. This visualization reveals distinct energy patterns between these classes, a consistency also found in sequence-based analyses. To explore structural information at lower hierarchical levels of SCOP, two folds (a.100 and a.104) from the all-alpha class, two superfamilies (a.29.2 and a.29.3) from fold a.29, and two families (a.25.1.0 and a.25.1.2) from superfamily a.25.1 were randomly selected. Figure 3 displays two figures per panel, with the left figure illustrating CPE profiles and the right figure showcasing SPE profiles. UMAP plots in Fig. 3 demonstrate that

protein domains within the same fold, superfamily, or family share similar energy patterns and cluster together.

To delve deeper into differences in distances among protein domains within the same class, we calculated pairwise distances for domains within the all-alpha class from the ASTRAL95 dataset. Subsequently, these distances were compared with distances from domains across different classes. As shown in Supplementary Fig. 4A, B, intraclass distances in purple are significantly lower than interclass distances in yellow. Similar results were obtained when calculating pairwise distances from domains within fold a.29 and comparing them with distances from domains in different folds within the all-alpha class. Likewise, distances between energy patterns of domains within the same superfamily a.29.3 are significantly less than distances between energy patterns of domains within fold a.29 that belong to different superfamilies (Supplementary Fig. 4C, D). Consequently, it can be inferred that energy patterns of domains belonging to the same superfamily/fold/class exhibit higher similarity than those from different superfamilies/folds/classes.

It is commonly assumed that proteins sharing similar structures also exhibit similar functions. Several measurements have been developed to assess protein structure similarity, each offering unique insights. Root Mean Square Deviation (RMSD)[24] quantifies the average spatial variance between corresponding atoms or components within superimposed proteins, providing a fundamental measure of structural deviation. For our analysis, RMSD calculations were performed by superimposing corresponding atomic coordinates using a least-squares fitting procedure implemented in R software, focusing on backbone Cα atoms to assess the overall fold without the influence of

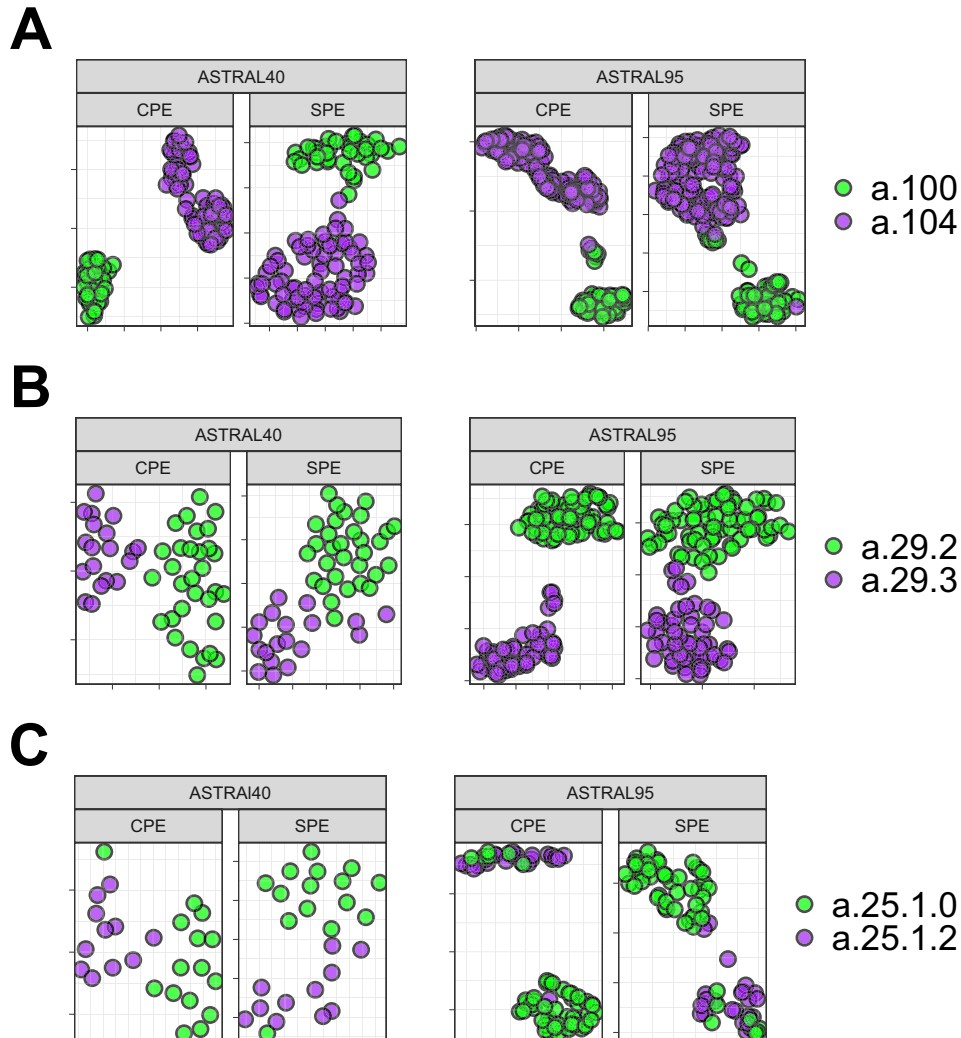

**Fig. 3 | UMAP Visualization of Energy Profiles.** The UMAP projection of structural profile of energy (SPE) and Compositional Energy Profiles (CPE) of protein domains from ASTRAL40 and ASTRAL95 represents the structural information embedded in energy profiles across hierarchical levels of SCOP; each panel includes two figures, one generated by CPE (left panel) and the other by SPE (right panel), revealing that protein domains sharing the same (**A**) fold, (**B**) superfamily, and (**C**) family exhibit comparable energy profile patterns. The folds a.100 and a.104, superfamilies a.29.2 and a.29.3, as well as families a.25.1.0 and a.25.1.2, are randomly selected for analysis, and the UMAP plots were generated using parameters n_neighbors = 30 and min_dist = 0.1. Source data are provided as a Source Data file.

side-chain orientations. While RMSD is widely used, it heavily relies on direct spatial overlap and is sensitive to outlier regions. This sensitivity often penalizes flexible regions or domain movements, such as hinge motions or flexible loops, potentially obscuring meaningful similarities in overall protein fold when local variations are present. The TM-score (Template Modeling score)[25] evaluates similarity by considering both residue-level alignment and overall topology, offering a nuanced assessment of structural resemblance. Unlike RMSD, TM-Score is less sensitive to domain-level movements but has its own limitations. Specifically, when comparing proteins with highly variable structures, TM-Score tends to favor the alignment of larger structural elements, potentially leading to lower scores for proteins with significant domain rearrangements despite sharing similar overall folds. Additionally, TM-Score may not adequately account for functional sites that involve small but critical local structural differences. TM-Vec[26], a recent advancement, employs deep learning techniques trained on diverse protein structures to enhance accuracy and efficiency in similarity assessment. TM-Vec maps protein structures into a vector space, allowing comparisons based on the distances between their vector representations. While highly accurate in detecting remote homology and structural similarities, TM-Vec's reliance on training data introduces inherent biases. These biases are particularly evident when evaluating proteins with rare or novel folds that are underrepresented in the training set. Furthermore, as a black-box model, TM-Vec offers limited interpretability regarding the specific structural features contributing to the similarity scores, which can be a limitation when detailed structural insights are required. On the alignment front, we utilized GR-Align[27] with its default parameters, employing the graphlet degree similarity (GDS) metric to capture topological similarities between protein structures. The GDS metric compares the distributions of small-connected subgraphs (graphlets) within the protein structures. GR-Align is robust in identifying proteins with similar topological arrangements, but its sensitivity to minor structural variations can lead to higher false positives when comparing proteins with subtle differences in their tertiary structures. Additionally, GR-Align does not incorporate sequence information or account for conformational flexibility, which may limit its ability to discern functionally relevant structural variations, particularly those involving dynamic regions of proteins. Finally, the Hausdorff distance[28] provides a measure of dissimilarity between sets of points, offering further insight

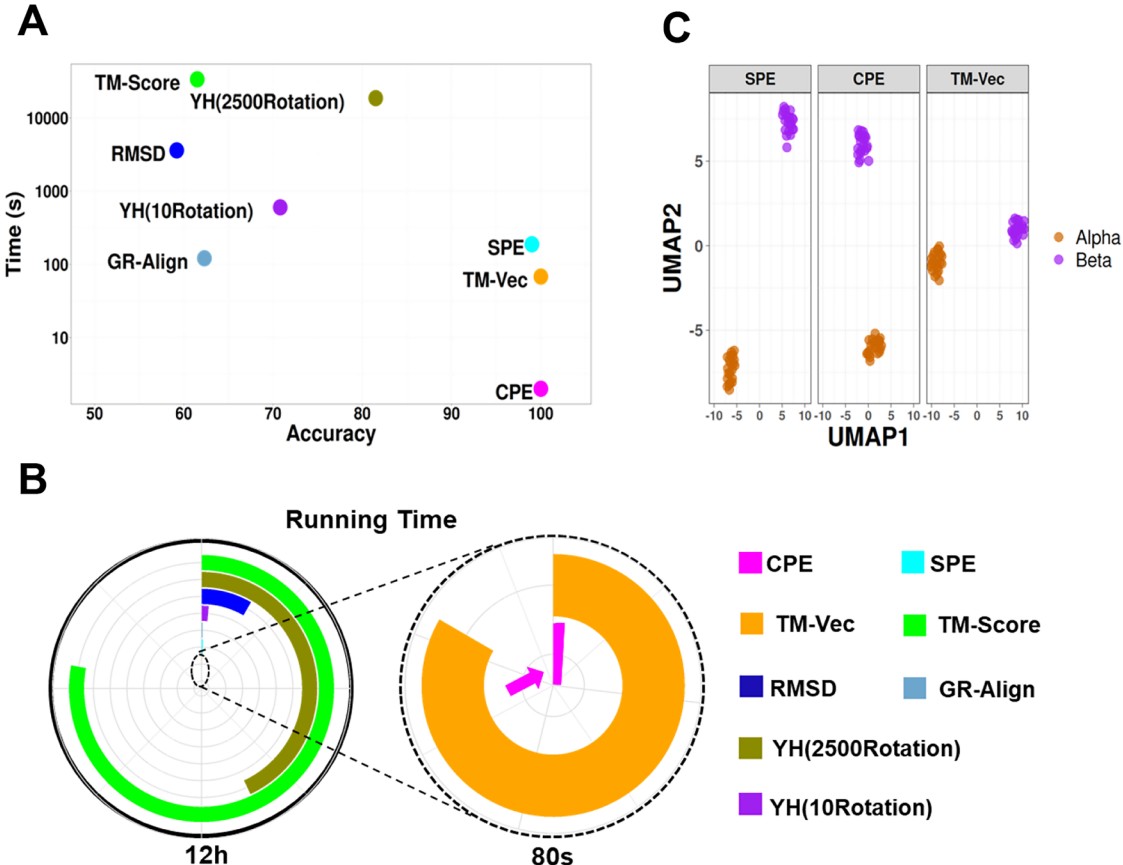

**Fig. 4 | Performance and Computational Efficiency of Protein Dissimilarity Measures. A** Time versus accuracy for the 1-NN classifier using GR-Align, RMSD, TM-score, Yau-Hausdorff distance, TM-Vec, and the distance between energy profiles SPE and CPE as measures of protein dissimilarity. **B** Running times of the evaluated methods, scaled to 12 h, with an inset zooming in on the region indicated by the dashed circle. The entire circle represents 80 s. Each method is represented by different colors as indicated in the figure legend. **C** The UMAP projection of α and β globins from the hemoglobin biological unit using CPE, SPE, and TM-Vec representations. n_neighbors = 13, min_dist = 0.1. Source data are provided as a Source Data file.

into structural comparisons. In our study, the Yau-Hausdorff distance was calculated by comparing point sets representing the protein structures, specifically using distances from the structural centroids to the Cα atoms. This method captures overall geometric differences between structures by measuring the maximal deviation between point sets in a bidirectional manner. However, it may not fully account for local structural variations or conformational changes, such as those occurring at active sites or ligand-binding regions, which are crucial for functional similarity. Moreover, the method assumes that global geometric similarity correlates with functional similarity, which may not always hold true, especially for proteins whose function is dictated by specific local conformations.

We employed a benchmark dataset from the CATH v4.2.0 database, consisting of 251 protein domains from two distinct protein families: the C-terminal domain in the DNA helicase RuvA subunit (representing the Alpha class, characterized by Orthogonal Bundle Architecture, Helicase, and Ruva Protein fold, with CATH Code: 1.10.8.10), and the Homing endonucleases (belonging to the Alpha and Beta class, featuring Roll Architecture, and Endonuclease I-creI fold, with CATH Code: 3.10.28.10). The protein domains varied in the number of residues, ranging from 44 to 854, with an average of 211. Proteins were classified using the 1-NN (1-nearest neighbor) method based on various similarity metrics, including GR-Align, RMSD, TM-score, Yau-Hausdorff distance, TM-Vec, and energy profile distances (CPE and SPE). As shown in Supplementary Table 1 and Fig. 4A, B, both CPE and TM-Vec achieved 100% accuracy in distinguishing between

the two protein families. However, the CPE method significantly reduced computation time, completing the task in just one second compared to TM-Vec's 67 s. Furthermore, CPE, SPE, and TM-Vec outperformed GR-Align, RMSD, TM-score, and YH in terms of accuracy. Notably, our approach eliminates the need for structural alignment by relying on energy profile calculations and distance measurements. The computation times listed in Supplementary Table 1 were measured on a system with a 2.4 GHz processor and 8GB of RAM.

To evaluate the effectiveness of energy profiles in protein superfamily classification, we examined five distinct SCOP superfamilies: winged helix (a.4.5), PH domain-like (b.55.1), NTF-like (d.17.4), Ubiquitin-like (d.15.1), and Immunoglobulins (b.1.1)[29]. Our classification approach utilized both energetic profiles and TM-Vec representations as features, applying 1-nearest neighbor (1-NN) classifier. The results, summarized in Table 1, report metrics for accuracy and F1-score, demonstrating the model's effectiveness. All methods achieved performance levels approaching 100%, as shown in Table 1. The CPE method, in particular, not only matched TM-Vec in terms of accuracy but also exhibited faster computation times.

To further validate our approach, we also analyzed the two subfamilies, α and β globins, which are part of the hemoglobin biological unit. Although α and β globins are closely related and share a common evolutionary origin, they have distinct and well-characterized functions within the hemoglobin α2β2 tetramer, despite their highly similar structures. Freiberger et al.[23] utilized a non-redundant set of experimental structures representing 21 mammalian hemoglobins (both α-

**Table 1 | Total accuracy and F1 measure for each of the five superfamilies by 1-NN based on CPE, SPE, and TM-Vec**

| Method | Time (Sec) | Accuracy | F1 Measure | | | | |
|---|---|---|---|---|---|---|---|
| | | | wigend_helix | PH.domain-like | NTF-like | Ubiquitin-like | Immunoglobulins |
| CPE | 103 | 0.98 | 0.98 | 0.96 | 0.99 | 0.99 | 0.99 |
| SPE | 3524 | 0.98 | 0.98 | 0.96 | 0.99 | 0.98 | 0.98 |
| TM_Vec | 955 | 0.99 | 0.99 | 1 | 1 | 0.99 | 0.99 |

and β-globins), and their analysis revealed distinct patterns of conserved energetic frustration, corresponding to their divergent functional roles in hemoglobin. Notably, specific residues exhibited highly frustrated interactions in one family while maintaining stability in the other, indicating evolutionary adaptations that reflect the distinct structural and functional demands of each globin subunit. We computed the CPE, SPE, and TM-Vec representations for this dataset. Figure 4C presents the UMAP projection for these proteins, where all methods effectively differentiate between the α and β globins.

## Phylogeny inference of the ferritin-like superfamily

In conjunction with the organizational frameworks provided by SCOP, CATH, and Pfam for the protein universe, it is important to note their limitations, as they may present conflicting classifications and lack the ability to elucidate evolutionary relationships between individual superfamilies across long evolutionary distances. Lundin et al.[19] conducted a comprehensive analysis of protein structures within the functionally diverse ferritin-like superfamily and investigated how ferritin-like proteins are classified across Pfam, SCOP, and CATH. Notably, this superfamily encompasses a diverse range of proteins, including iron-storing ferritins, methane monooxygenases, the small subunit of Ribonucleotide reductase-like (RNR R2), rubrerythrins, bacterioferritins, Dps (DNA binding protein from starved cells that protects against oxidative DNA damage), and Dps-like proteins. As discussed by Lundin et al.[19] at the superfamily level, the classification of the ferritin-like superfamily appears consistent across these databases but does differ in the amount of information provided regarding the relationships and functions of superfamily constituents. So, although the classification in all three databases is hierarchical, they do not encompass all level of functional and evolutionary information. To address this, they employed an evolutionary network construction approach to unveil relationships among proteins beyond the twilight zone, where sequence similarity alone fails to facilitate meaningful evolutionary analysis. The low sequence similarities across this superfamily make it feasible to construct sequence-based phylogenies only for specific subsets. Consequently, addressing this challenge requires efforts to integrate structural information with sequence-based phylogenies. Malik et al.[30], and Puente-Lelievre et al.[31] delved into the evolutionary relationships of this superfamily by creating a phylogenetic network. They employed the distance-based Neighbor-Net network method[32], utilizing distances calculated through structure-based alignment methods. Figure 5A depicts the schematic tree built by Malik et al.[30], and Lelievre et al.[31].

Using the same protein structures from this superfamily as employed by Malik et al.[30] and Lelievre et al.[31], we reconstructed the phylogeny of these proteins based on energy profiles (CPE and SPE) and the TM-Vec method. The dataset focuses on the SCOP Ferritin-like superfamily (a.25.1), which includes two curated families: Ferritin (a.25.1.1), containing ferritins, bacterioferritins, and Dps proteins, and Ribonucleotide Reductase-like [RNR] (a.25.1.2), comprising the RNR R2 subunit, BMM, and fatty acid desaturases. Phylogenetic trees constructed from SPE, TM-Vec, and CPE data using the phangorn package[33] and visualized with SplitTree software[34] are shown in Fig. 5B, D, F.

In line with the reference in Fig. 5A, a notable observation is that all three phylogenetic trees exhibit two main branches (marked by dashed lines), representing the two families a.25.1.1 and a.25.1.2. This outcome highlights the effectiveness of the methods in accurately differentiating the superfamily into its respective families. Delving into specifics, the family a.25.1.1 (depicted by orange color triangles) further divides into four subgroups: ferritins, Dps, Rubrerythrin, and Bacterioferritins indicated by distinct colors in Fig. 5A. On the other hand, the second branch related to the a.25.1.2 family (dark blue triangles), despite SCOP and CATH assigning these proteins to a unified RNR-like family, reveals three distinct families according to Pfam—Phenol_Hydrox (PF02332), Ribonuc_red_sm (PF00268), and Fatty acid desaturase (PF03405). The protein groupings presented by Lelievre et al.[31] in Fig. 5A are color-coded, corresponding to the colors used in Fig. 5B–G.

For a comprehensive comparison with the manual annotation in Fig. 5A, we calculated the average inter-protein distances across the eight subfamily groups using three distinct methods: CPE, SPE, and TM-Vec. These distance matrices were used to construct neighbor-joining phylogenetic trees, as shown in Fig. 5C, E, G. To assess the accuracy of the subfamily arrangements, we compared the inferred phylogenetic trees with the manually annotated subfamily relationships, with color-coding corresponding to subgroups in Fig. 5A. The order of subgroups in the manually annotated network was then compared with the order produced by SPE, TM-Vec, and CPE (Fig. 5C, E, G). We found that while TM-Vec performs well in clustering similar proteins, it struggles to accurately reflect the evolutionary relationships between subfamilies. For instance, in the a.25.1.2 family, TM-Vec incorrectly places the subgroup BMM-b first, whereas it should appear last in the manual network. Similarly, in the a.25.1.1 family, although TM-Vec correctly identifies the initial branch, it misorders the remaining subgroups. To quantify these deviations, we calculated the Spearman correlation between the predicted branching orders from each method and the manual network. These correlations, summarized in Table 2, demonstrate that both SPE and CPE outperform TM-Vec in aligning with the manually annotated network. Notably, SPE achieves a perfect correlation in the a.25.1.1 family, while TM-Vec shows a negative correlation in the a.25.1.2 family, illustrating its limitations. Our findings indicate that the energy-based phylogenies within the ferritin-like superfamily reveal substantial relationships among its members, aligning closely with established evolutionary connections and functional roles suggested by Malik et al.[30], and Lelievre et al.[31] showed in Fig. 5A.

## Clustering of the SARS-CoV-2, SARS-CoV and 2012 MERS-CoV proteins

Over the past two decades, coronaviruses (CoVs) have been linked to several significant outbreaks, including the SARS-CoV outbreak in 2002-2003, the MERS-CoV incident in 2012, and the recent COVID-19 pandemic caused by SARS-CoV-2 in late 2019. Since February 2020, a substantial number of SARS-CoV-2 protein structures have been deposited in the Protein Data Bank (PDB), with the spike glycoprotein being of particular interest due to its crucial role in viral infection by mediating host receptor binding. This protein is a primary target for neutralizing antibodies and vaccine development.

To explore the structural landscape and evolutionary relationships of these spike glycoproteins, we used the CoV3D database(https://cov3d.ibbr.umd.edu), which provides a comprehensive

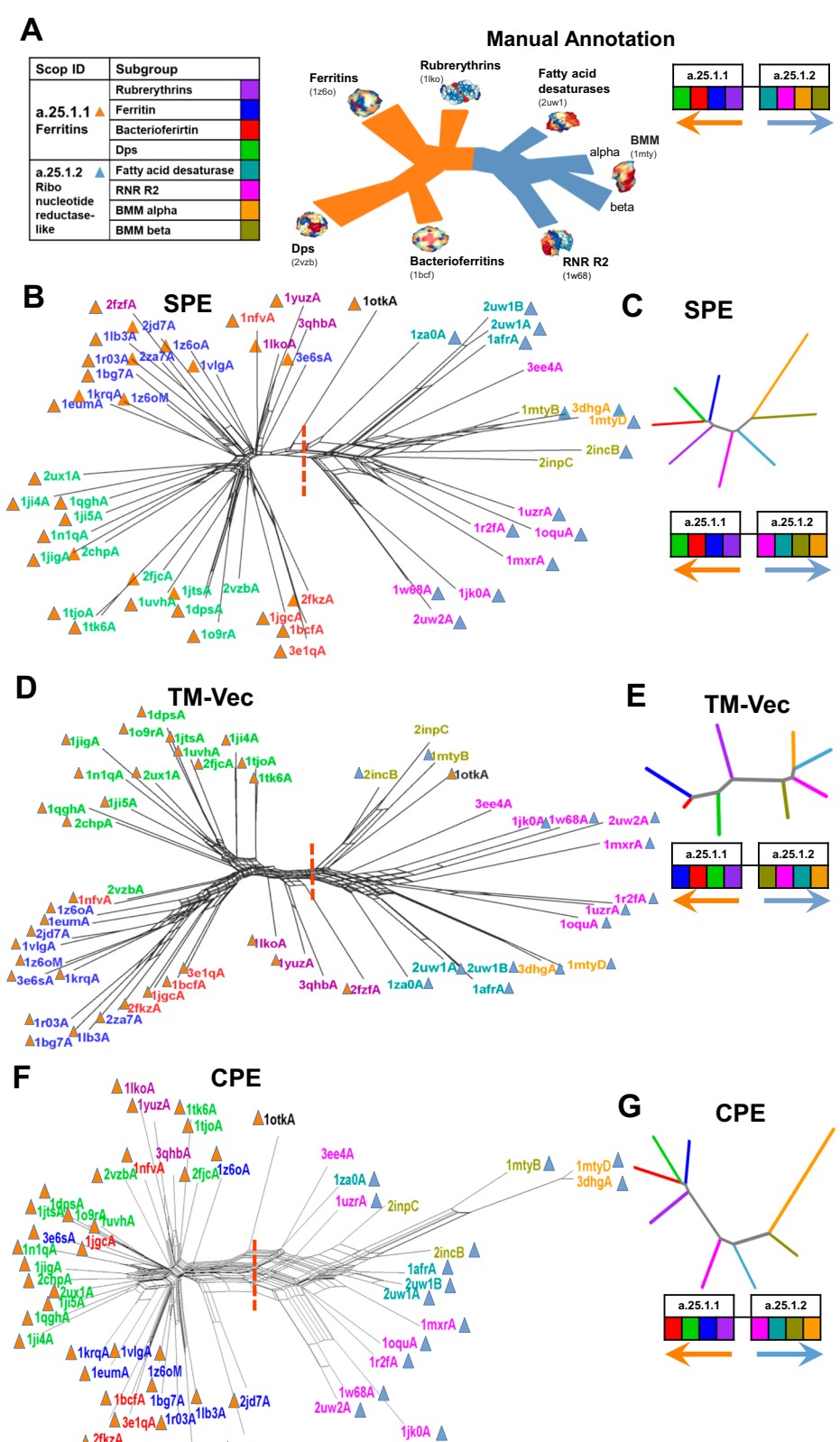

**Fig. 5 | Phylogenetic network reconstruction of the ferritin-like superfamily.**
**A** Schematic representation of the relationships among major ferritin-like protein families, with each subgroup shown in a distinct color. **B** Phylogenetic network reconstructed using SPE. **C** Neighbor-joining tree generated based on the average distances between subgroups using SPE. **D** Phylogenetic network reconstructed using TM-Vec. **E** Neighbor-joining tree generated using the average distances between subgroups with TM-Vec. **F** Phylogenetic network reconstructed using CPE.

**G** Neighbor-joining tree based on the average distances between subgroups using CPE. The red dotted line highlights the separation between two SCOP families: ferritins (SCOP ID a.25.1.1), which includes the Bacterioferritin, Ferritins, Dps, and Rubrerythrin subgroups, and the Ribonucleotide Reductase-like family (SCOP ID a.25.1.2), which includes the BMM-alpha, BMM-beta, Fatty_acid, and RNRR2 subgroups. The inferred arrangement of subfamilies using each method is shown to the right of (**C**, **E**, **G**). Source data are provided as a Source Data file.

collection of coronavirus protein structures and their interactions with antibodies, receptors, and small molecules[20]. From this resource, we curated a dataset of 143 spike glycoprotein structures, all containing a closed receptor-binding domain (RBD). This dataset comprises 80 spike protein chains from SARS-CoV-2, 31 from SARS-CoV, and 32 from MERS-CoV (Supplementary Table 4).

Initially, we conducted a multiple sequence alignment of spike proteins using the ClustalW method via the msa package[35] in R. Protein distances were calculated with the seqinr package[36], using sequence identity as the distance metric. Phylogenetic trees were subsequently generated using the UPGMA method from the phangorn package[33], and the resulting tree based on sequence similarity is presented in Fig. 6A. To further examine structural variations and relationships among the spike glycoproteins, we applied structural methods including RMSD, TM-Score, and SPE, alongside sequence-based

methods such as CPE and TM-Vec, to calculate pairwise distances and cluster the spike glycoproteins into three distinct groups corresponding to SARS-CoV, MERS-CoV, and SARS-CoV-2. This clustering offers a visual representation of the structural and evolutionary relationships within this protein family, as depicted in Fig. 6A–F.

All methods consistently showed a clear separation between the SARS-CoV/SARS-CoV-2 lineage and MERS-CoV, which belongs to a different coronavirus subgenus. However, as illustrated in Fig. 6C, E, and F, methods like RMSD, TM-Score, and TM-Vec were less effective in capturing these evolutionary patterns. For instance, certain SARS-CoV proteins were misclassified and appeared distant from their respective groups (highlighted by orange circles in Fig. 6E). Similarly, the TM-Vec method misclassified some SARS-CoV proteins and failed to accurately group MERS-CoV proteins (Fig. 6C). Additionally, we performed a bootstrap analysis with 100 replicates to assess the robustness of phylogenetic tree reconstructions using CPE, SPE, TM-Vec, and MSA methods. Confidence intervals were calculated to statistically validate the accuracy of these results. The bootstrap values and confidence intervals for the main branches distinguishing the three species are provided in Fig. 6A, B, D, reinforcing the reliability of the energy profile-based model in reconstructing the phylogenetic tree. Detailed bootstrap results for all branches are available in Supplementary Figs. 5–8.

**Table 2 | Spearman Correlation Between the Manual Network and Predicted Branching Orders by SPE, TM-Vec, and CPE**

| Method | a.25.1.1 | a.25.1.2 |
|---|---|---|
| CPE | 0.8 | 0.6 |
| SPE | 1 | 0.6 |
| TM_Vec | 0.2 | −0.4 |

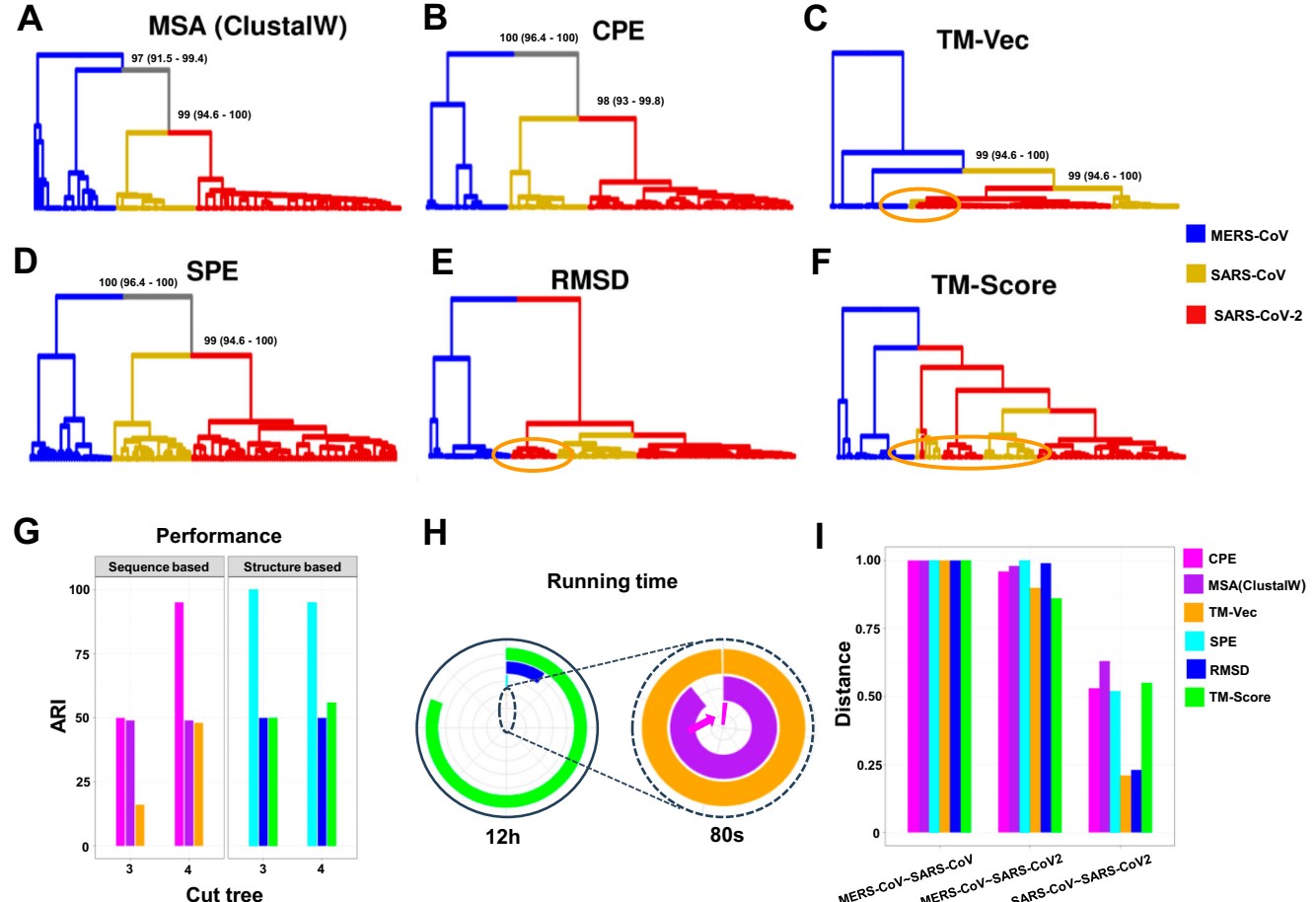

**Fig. 6 | Clustering analysis of spike glycoprotein structures from SARS-CoV, SARS-CoV-2, and MERS-CoV.** The dendrograms depict the clustering of spike glycoprotein structures from the three viruses: SARS-CoV, SARS-CoV-2, and MERS-CoV. The clustering is based on pairwise distances calculated from different methods: (**A**) protein sequence, (**B**) CPE, (**C**) TM-Vec, (**D**) SPE, (**E**) RMSD, and (**F**) TM-Score. The leaves of each tree are color-coded to indicate the originating virus for each spike glycoprotein structure. **G** Displays the ARI values for each method, (**H**) shows the running time associated with each method scaled to 12 h, with an inset zooming in on the region indicated by the dashed circle. The entire circle represents 80 s, and (**I**) presents the average distance between the three virus groups as calculated by each method. Source data are provided as a Source Data file.

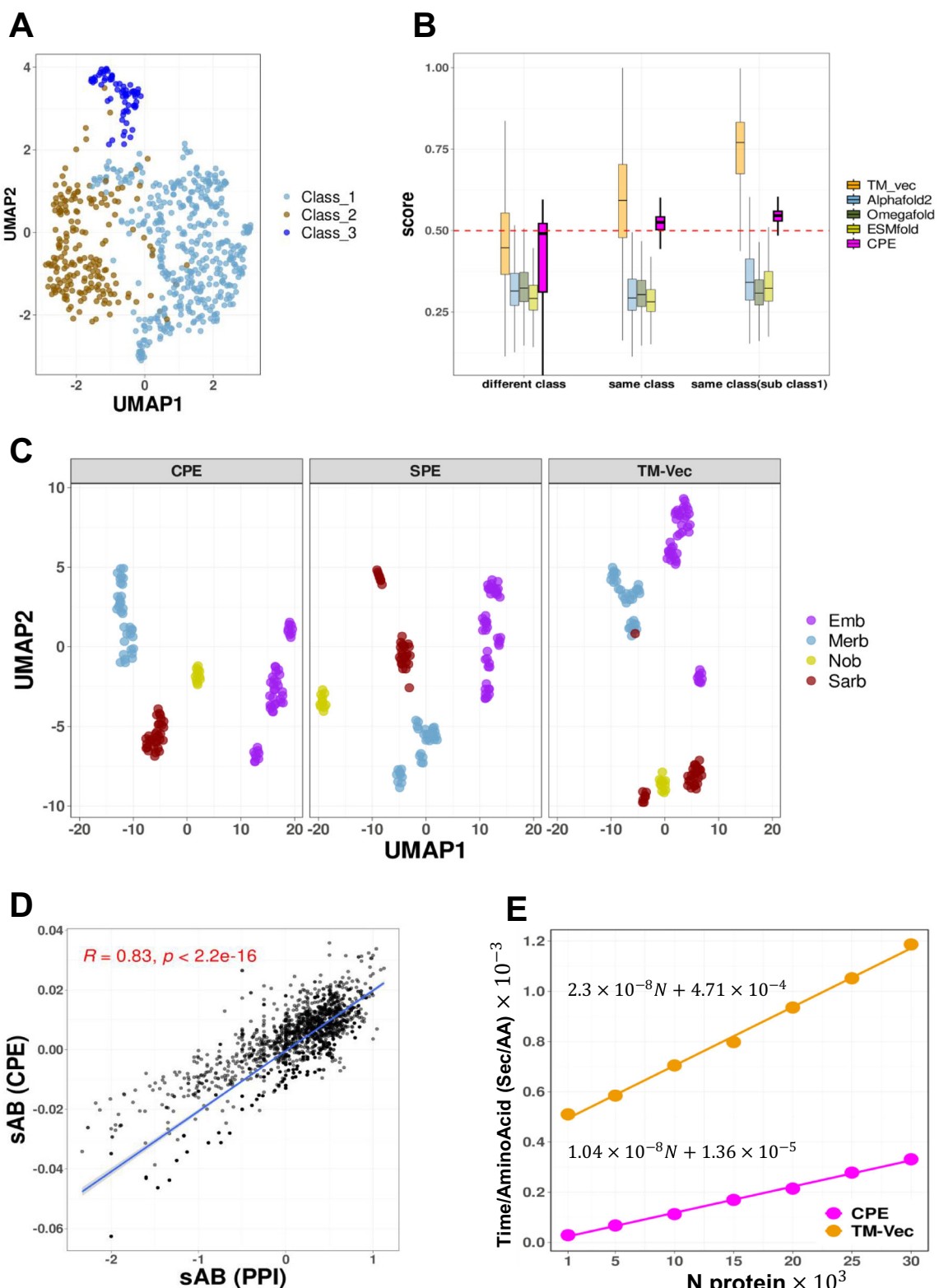

In terms of computational efficiency, our methods—CPE and SPE—are significantly faster than the alternatives. CPE completes the analysis in 0.9 s, and SPE takes just 3 min, whereas TM-Vec requires 89 s, RMSD takes 70 min, MSA takes 72 s, and TM-Score takes 9.7 h (Fig. 6H). This demonstrates the computational advantage of our methods, making them both accurate and efficient for clustering analyses. To

evaluate clustering performance, we used the Adjusted Rand Index (ARI), which measures the similarity between two clustering results, with values ranging from −1 to 1 (where 1 indicates perfect agreement). As shown in Fig. 6G and Supplementary Table 3, CPE method achieved the highest clustering performance with an ARI of 0.95 at a cut tree of 4, while TM-Vec performed best at a cut tree of 5, with an ARI of 0.87

**Fig. 7 | UMAP Projection of Energy Profiles for Bacteriocins and Betacoronavirus Domains, with Distance Comparisons and Scalability Analysis.**
**A** UMAP projection of Compositional Energy Profiles (CPE) for 690 peptides, representing three different classes of bacteriocins. **B** Comparison of CPE distances with the TM-scores produced by running TM-align on structures predicted by AlphaFold2, OmegaFold and ESMFold, and TM-Vec for all pairs of bacteriocins, pairs at different classes ($n = 125869$), pairs at the same class ($n = 111349$), pairs from the same class from subclass1 ($n = 13431$). Statistics assessed by two-tailed Student's t-test. Boxplots display the median (center line), the 25th and 75th percentiles (bounds of the box), and the minimum and maximum values (whiskers) excluding outliers. Comparisons of CPE distances revealed statistically significant differences across the groups (different class, same class, and subclass1 within the same class), with p-values for all three tests being <10e−16. The exact p-value is less than 10e-16,

which is below the precision threshold of standard statistical computations. CPE distances are normalized by min-max normalization. **C** Papain-like Protease (PLPro) domains across Betacoronavirus subgenera. The UMAP projection shows clustering of PLPro domains from Sarbecovirus ($n = 31$), Nobecovirus ($n = 11$), Merbecovirus ($n = 34$), and Embecovirus ($n = 45$) using SPE, CPE, and TMVec representations. n_neighbors = 13, min_dist = 0.5. **D** Two-sided Pearson correlation comparing Protein-Protein Interaction Network Distances (X-axis) with Energy Profile Distances (Y-axis). The blue line illustrates the least squares regression line, and the gray shaded area represents the 95% confidence intervals around the regression line. Each point corresponds to a drug combination. R indicates the correlation coefficient, and p shows the p-value. **E** Scalability of CPE and TM-Vec. Processing time per amino acid for subsets from the ASTRAL95 dataset, ranging in size from 1000 to 30,000 proteins (at intervals of 5000). Source data are provided as a Source Data file.

and MSA with cut tree of 3 with ARI of 0.49. Among the structure-based methods, SPE achieved a perfect ARI of 1 at a cut tree of 3, while RMSD and TM-Score performed best at cut trees of 6 and 4, with ARIs of 0.73 and 0.56, respectively (Supplementary Table 2). These results highlight the robustness of our methods for both sequence- and structure-based clustering. We also computed the average distance between the three virus groups—SARS-CoV, SARS-CoV-2, and MERS-CoV—across all six methods. All methods consistently show that SARS-CoV-2 is more closely related to SARS-CoV than to MERS-CoV (Fig. 6I). Additionally, the distance between SARS-CoV-2 and MERS-CoV is nearly identical to the distance between SARS-CoV and MERS-CoV.

## Clustering of bacteriocins

Bacteriocins are peptides produced by bacteria that act as strong antibacterial agents against other, typically closely related microbial species. We analyzed the bacteriocins family available in the BAGEL database, those with a length larger than 30 amino acids, including a total of 690 proteins[21]. Detecting and understanding these peptides is crucial due to their ecological importance, but their diverse sequences and structures make them challenging to identify using traditional methods. To address this issue, the BAGEL tool was developed in 2006, specifically designed for identifying Ribosomally synthesized and post-translationally modified peptides (RiPP) and bacteriocin biosynthetic gene clusters (BGCs). BAGEL categorizes bacteriocins based on size and stability into RiPPs (also defined as class I bacteriocins by BAGEL), class II bacteriocins (small heat stable proteins <10 kDa) and class III bacteriocins (large heat-labile proteins > 10 kDa). As shown in Fig. 7A, our analysis revealed that profile of energy (CPE) can clearly partition bacteriocins according to BAGEL annotation. Hamamsy et al.[26] leveraged the deep protein language models to develop the TM-Vec model, which is trained on pairs of protein sequences and their TM-scores. We compared CPE distances to the TM-scores of protein structures predicted by AlphaFold2[37], OmegaFold[38], and ESMFold[39], as well as the TM-Vec predicted by the model. As demonstrated in Fig. 7B, the TM-score of proteins predicted by AlphaFold2, OmegaFold, and ESMFold from the same class is similar to proteins from different classes. TM-Vec is effective at distinguishing between bacteriocins from the same class and proteins from different classes. Although there is some overlap between TM-Vec values from proteins from the same class and other classes. Our method also effectively distinguishes between proteins from the same class and those from other classes in bacteriocin dataset.

## Effective drug combination suggestion using energetic signatures

The identification and validation of effective drug combinations are crucial in the treatment of various complex diseases, aiming to enhance therapeutic efficacy while minimizing toxicity[40,41]. However, this task is hindered by a combinatorial explosion resulting from the

multitude of potential drug pairs. Cheng et al.[22] introduced a network-based methodology to pinpoint clinically effective drug combinations tailored to specific diseases. This approach involved assessing the network-based relationships among drug targets and disease proteins within the human protein-protein interactome. By quantifying these relationships, they identified clusters of drugs that exhibited correlations with therapeutic effects. The drugs within these clusters targeted the same disease module but belonged to separate neighborhoods. This innovative network methodology presented by Cheng et al. provides a generic and powerful means to discover effective combination therapies during drug development. Disease proteins were observed to form localized neighborhoods, referred to as disease modules, rather than being randomly distributed throughout the interactome. To characterize the mutual relationship between two drugs and a disease module, they employed the following network-based proximity measure:

$$s_{AB} = <d_{AB}> - \frac{<d_{AA}> + <d_{BB}>}{2} \tag{1}$$

This measure assessed the network proximity of drug-target modules A and B by comparing the mean shortest distance within the interactome between the targets of each drug ($<d_{AA}>$ and $<d_{BB}>$) to the mean shortest distance between A and B target pairs $<d_{AB}>$. When $s_{AB} < 0$, the targets of the two drugs are in the same network neighborhood; when $s_{AB} > 0$, the two targets are topologically separated.

The authors demonstrated that the topological relationship between two drug-target modules, as indicated by $s_{AB}$, reflects both biological and pharmacological relationships. They also showed that the network proximity ($s_{AB}$) of drug-drug pairs in the human interactome correlates with chemical, biological, functional, and clinical similarities. This led them to conclude that each drug-target module possesses a well-defined network-based footprint. If the footprints of two drug-target modules are topologically separated, the drugs are considered pharmacologically distinct. Conversely, if the footprints overlap, the magnitude of the overlap indicates the strength of their pharmacological relationship. A closer network proximity of targets in a drug pair suggests higher similarities in their chemical, biological, functional, and clinical profiles.

Here, we used the following separation measure, denoted by $E_{AB}$, based on similarity between profiles of energies of protein targets:

$$E_{AB} = <d_{AB}> - \frac{<d_{AA}> + <d_{BB}>}{2} \tag{2}$$

where,

$$<d_{AB}> = \frac{1}{||A|| + ||B||} \left( \sum_{a \in A} \min_{b \in B} d(a, b) + \sum_{b \in B} \min_{a \in A} d(a, b) \right)$$

and $d(a, b)$ represents the Manhattan distance between the energy profiles of proteins $a$ and $b$.

Figure 7D depicts the correlation between $s_{AB}$ values, as computed by Cheng et al.[22], for a set of 65 antihypertensive drugs exhibiting complementary exposure to the hypertension disease module, and the corresponding $E_{AB}$. The results demonstrate a strong correlation between $s_{AB}$ and $E_{AB}$, suggesting that the energy profile holds promise for predicting drug combinations. It is important that our approach only requires protein sequences and is significantly faster than computing the shortest path in a protein-protein interaction network.

**Large-scale application of family detection in coronaviruses**
To evaluate our method on a larger dataset, we utilized a coronavirus dataset that was previously generated and analyzed by Freiberger et al.[23] Initially, they retrieved homologous sequences from the SARS-CoV-2 reference genome MN985325, with low-quality and non-Coronaviridae sequences excluded[42]. Then sequences were aligned using MAFFT software[43], with non-structural protein sequences trimmed to focus on regions specifically relevant to SARS-CoV-2. These aligned sequences were then clustered using CD-Hit[44] to ensure high similarity within each cluster. Structural models for these sequences were generated using AlphaFold2, retaining only those that met stringent quality criteria. The high-quality models were subsequently grouped into subfamilies using S3Det software[45], allowing for the identification of Specificity Determining Positions (SDPs) within the proteins. This meticulous process resulted in a final set of 28 high-quality protein families, comprising 4405 protein models. For each protein in this dataset, we computed the energy profiles CPE and SPE, along with the distance between these profiles and the cosine similarity between pairs of TM-Vec representations. The 1-nearest neighbor (1-NN) method was then utilized to classify the proteins into different families. The results, shown in Table 3, include metrics for accuracy and F1-score, demonstrating the effectiveness of our model. Both methods achieved performance levels near 100%, with CPE offering faster performance. Detailed outcomes of the 1-NN classification are provided in Supplementary Supplementary Tables 5–7, and the UMAP projections of SPE, CPE, and TM-Vec representations are displayed in Supplementary Figs. 9–11.

The Papain-like Protease (PLPro) domain plays a crucial role in viral replication by catalyzing the proteolysis of viral polyproteins. In addition, PLPro interacts with two host proteins, ubiquitin (Ub) and the ubiquitin-like interferon-stimulated gene 15 protein (ISG15), allowing the virus to evade or weaken the host immune response. In this dataset, PLPro was divided into four subfamilies aligned with the Betacoronavirus subgenera: Sarbecovirus ($n = 31$), Nobecovirus ($n = 11$), Merbecovirus ($n = 34$), and Embecovirus ($n = 45$)[23]. The UMAP representation for both CPE and SPE is shown in Fig. 7C, where proteins from each subfamily are clearly clustered together.

## Discussion
The continuous growth of protein databases highlights the importance of understanding their functional characteristics. It's widely recognized that proteins with similar structures often perform similar

functions. Additionally, there's a common belief that proteins with similar structures also share similar energy levels. Therefore, our study aims to pioneer a approach by directly linking protein energy landscapes to their functional attributes. By investigating this relationship, we seek to uncover insights into how protein structure, energetics, and biological activity are interconnected. Knowledge-based potentials are energy functions derived from known protein structures. In our study, we developed a potential function to calculate the energy between pairs of amino acids, generating energy profiles based on both sequence and three-dimensional structure. A significant achievement of our study is the high correlation observed between energy estimates derived from sequence and those from structural data, allowing for the derivation of energy profiles based solely on sequence information, which enables fast and accurate computational analysis. However, it's worth noting that the reliance on knowledge-based potentials is dependent on known protein structures, potentially limiting the generalizability of results to proteins with varied structural characteristics or those are underrepresented in existing databases. Furthermore, despite the promising correlation between energy estimates derived from sequence and structural data, it is possible that there are complexities in accurately capturing the entirety of protein energetics solely from sequence information, which could affect the reliability of the resulting energy profiles. To address these issues, one possible option is to adjust the energy profile, such as through reweighting, to specific applications, such as protein remote homology detection or drug-target affinity prediction.

We employed Uniform Manifold Approximation and Projection (UMAP) to visualize energy profiles at both sequence and structural levels derived from protein domains within the ASTRAL database, revealing their capacity to distinguish proteins across various hierarchical levels, including class, fold, superfamily, and family. Notably, the Manhattan distance between energy profiles serves as a measure of dissimilarity, eliminating the necessity for structural or sequence alignment in protein comparison and resulting in significantly faster computational analyses, as demonstrated in Supplementary Table 1. The comparison table highlights notable differences in both accuracy and computational efficiency among the methods evaluated. The profile of energy (CPE) method demonstrates a remarkable accuracy of 100%, significantly surpassing other methods such as GR-Align, RMSD, and TM-Score, which range from 59.2% to 81.5%. This indicates that the CPE method excels in accurately distinguishing between protein structures at different superfamilies, showcasing its superiority in capturing structural dissimilarities effectively. In terms of computational efficiency, the CPE method stands out as the most time-efficient, requiring a mere 1 s for processing. In contrast, traditional methods like RMSD and TM-Score demand significantly longer computational times, ranging from 1 h to over 9 h. This stark difference underscores the efficiency of the CPE method, particularly in time-sensitive scenarios or large-scale protein structure comparison tasks.

Our method's efficacy was further assessed by comparing its results with TM-Vec method in classifying proteins across five distinct SCOP superfamilies, showcasing its superior computational efficiency. Particularly challenging is elucidating evolutionary relationships among superfamilies beyond the "twilight zone," where sequence similarity alone proves inadequate for meaningful analysis. To address this, we examined energy profiles to reconstruct a phylogenetic network of the Ferritin-like superfamily, incorporating proteins from the twilight zone. Our analysis, consistent with previous studies by Lundin et al.[19] and Malik et al.[30], unveiled substantial and valuable evolutionary signal preserved within energy profiles, indicating their potential as representative indicators of protein structure. Moreover, we examined the structural attributes of spike glycoproteins among three coronaviruses—SARS-CoV, MERS-CoV, and SARS-CoV-2—using a 210-dimensional energy profile combined with Manhattan distances. This study successfully grouped these proteins into specific clusters

**Table 3 | Large Scale: Analysis of the SARS-CoV-2 Proteome across 28 families, encompassing 4405 proteins. The overall accuracy, F1 score, and computation time for detecting families using the 1-NN classifier**

| Method | Time (Sec) | Accuracy | F1 Measure |
|--------|-----------|----------|------------|
| CPE | 49 | 0.9968 | 0.9946 |
| SPE | 2715 | 0.9973 | 0.9903 |
| TM_Vec | 685 | 0.9966 | 0.9925 |

corresponding to each virus, offering insights into their structural and evolutionary relationships. Additionally, our inquiry extended to 690 proteins within the bacteriocins family, encompassing various sizes and stability levels sourced from the BAGEL database. By employing the energy profile (CPE), we effectively distinguished bacteriocins according to BAGEL classifications, showcasing the usefulness of this method in protein classification, particularly in scenarios where proteins exhibit differing stabilities. Comparative analysis involving TM-scores from a range of prediction models emphasized the effectiveness of our approach in differentiating proteins within and across classes, thereby providing valuable insights into bacteriocins. In summary, our findings underscore the valuable insights offered by energy profiles across structural, functional, and evolutionary scales.

One of the significant applications of assessing protein similarity lies in quantifying the proximity between two drugs based on their protein targets. When the protein targets of two drugs exhibit similarity, it is reasonable to anticipate similarities in the drugs themselves. Our method, capable of quantifying the dissimilarity between two proteins, potentially encodes functional information that can be leveraged to gauge the similarity between two drugs according to their protein targets. Comparative analysis with a study conducted by Cheng et al.[22] demonstrates a notable correlation between our results, derived solely from protein sequence data, and theirs, obtained using protein-protein interaction data. It is worth reiterating that our method boasts remarkable speed compared to conventional approaches. By providing a rapid yet effective means of assessing protein similarity, our method offers promising implications for drug discovery and development, facilitating the identification of potential drug candidates with similar protein targets. This underscores the significance of leveraging computational methods to expedite drug discovery processes while maintaining robustness and accuracy. In conclusion, our research introduces the energy profile as an innovative feature set containing significant functional insights that can be utilized to represent proteins within machine learning methodologies for predicting protein function, drug-target interactions, and drug combination outcomes.

In our investigation, we examined the energy profile surrounding protein drug targets and demonstrated a strong correlation between our scoring system and that derived from protein-protein interaction networks. It's important to acknowledge that while a more sophisticated computational approach and experimental validation are crucial in drug combination study, these aspects fall beyond the purview of our manuscript.

Moreover, while our method bears significant implications for drug discovery and development, its efficacy might be limited by the availability and quality of protein sequence and structural data, as well as the inherent complexity of drug-target interactions. Therefore, it is imperative for independent research endeavors to address this crucial aspect and offer comprehensive insights into the practical application of our approach in real-world therapeutic contexts.

To evaluate the scalability of the CPE method, we performed an additional analysis by randomly selecting protein subsets from the ASTRAL95 dataset, ranging in size from 1000 to 30,000 proteins (at intervals of 5000). For each subset, we computed the pairwise distances between proteins using both the CPE and TM-Vec methods, while also recording the processing time per amino acid. This metric represents the total computation time divided by the cumulative number of amino acids across all analyzed protein domains. As shown in Fig. 7E, both methods demonstrate a linear increase in computation time per amino acid as the dataset size expands. However, the CPE method exhibits a gentler slope compared to TM-Vec, indicating superior scalability. These findings underscore the efficiency of the CPE method, particularly for handling large, complex datasets, making it highly suitable for high-throughput computational studies.

The CPE method leverages energy profiles derived from pairwise amino acid interactions, where each dimension of the energy vector corresponds to specific amino acid pairs. This structured, physically grounded representation makes the data more intuitive and interpretable because the calculated energy values directly reflect biologically meaningful aspects of protein interactions, such as stability, folding, and molecular dynamics. This clarity allows researchers to trace protein similarities back to the underlying energy landscapes, which are well-established in protein science. In contrast, TM-Vec utilizes deep learning embeddings that, although highly effective, function as a "black box." While TM-Vec can identify remote homologies and structural similarities, the embeddings it generates are abstract and difficult to deconstruct in a biologically meaningful way. This limits the ability to draw direct connections between the model's outputs and the physical or functional properties of proteins.

The CPE method, by focusing on energy profiles, offers distinct advantages in terms of interpretability. Energy profiles are inherently linked to protein folding, stability, and interaction networks, which are fundamental to biological function. For example, an increase in energy in specific pairwise interactions might suggest destabilizing mutations or conformational shifts that affect protein function. This explicit connection between energy and structural features allows for more transparent insights into how variations in energy impact the overall behavior and evolutionary relationships of proteins. By directly correlating these energy states with functional classifications such as folds, superfamilies, or evolutionary relationships, CPE provides clearer, actionable insights for researchers.

Additionally, because energy profiles can be tied to specific biophysical principles, CPE offers a mechanistic understanding of protein relationships that is often lacking in machine learning models like TM-Vec. In fields such as drug discovery or protein engineering, where understanding the precise molecular interactions is crucial, CPE provides a significant advantage in generating interpretable and actionable data. In conclusion, CPE's reliance on energy profiles provides not only a more interpretable but also a biophysically grounded model of protein similarity. This contrasts with TM-Vec's deep learning approach, which, while powerful, offers less transparency and explainability. CPE's approach is particularly valuable in contexts where understanding the biological and structural principles behind protein behavior is critical, such as in evolutionary studies, disease-related mutation analysis, and drug development.

## Methods

A curated dataset of non-redundant protein chains was utilized from PISCES[46]. The dataset was selected based on the following criteria:

- Pairwise sequence identity: Less than 50% to ensure non-redundancy.
- Resolution: Higher than 1.6 Å to guarantee structural accuracy.
- R-factor: Below 0.25 to ensure reliable crystallographic data.
- Protein length: Between 40 and 1000 residues to include proteins of varying sizes while excluding excessively short or long chains.
- Overlap: Proteins overlapping with the test sets from this manuscript were removed from the training set.

These filtered proteins were utilized to train and calculate the knowledge-based potential function as follows.

### Pairwise distance-dependent knowledge-based potential

Knowledge-based potentials are derived from databases of known protein structures and are essential for estimating the energies of pairwise interactions. These potentials can be based on various factors, including distance dependencies, dihedral angles, and accessible surface areas[8]. In this study, we employed a distance-dependent potential function where atomic contacts were identified using the tessellation method[9,47,48] as follows:

## Contact Identification

1. Representation: All amino acids in each protein chain were represented by their heavy atoms (excluding hydrogen atoms).

2. Delaunay Tessellation: A Delaunay tessellation of the resulting point set was computed using Qhull[49], identifying neighboring atoms based on spatial proximity.

3. Defining Contacts: Two atoms were considered to be in contact if they are connected by an edge in the Delaunay triangulation. This implies that they are not shielded from each other by other atoms, ensuring direct interaction without obstruction.

4. Distance Shells: The distances between contacting atoms were divided into 30 discrete shells, starting at 0.75 Å with each shell having a width of 0.5 Å. This binning allows for the extraction of distance-dependent interaction potentials.

Atom Types: A total of 167 atom types were considered by treating non-hydrogen atoms as distinct based on their specific amino acid residues.

Energy Calculation: The potential energy between two atoms $i$ and $j$ at distance $d$ was calculated using the following equation[7]:

$$\Delta E^{ij}(d) = RT\left[\ln\left(1 + M_{ij}\sigma\right) - \ln\left(1 + M_{ij}\sigma\left(\frac{f_{ij}(d)}{f_{xx}(d)}\right)\right)\right] \quad (3)$$

where $RT$ is constant and equal to 0.582 kcal/mole. $M_{ij}$ is the number of observations for atomic pair $i$ and $j$, $f_{ij}(d)$ is the relative frequency of occurrence for $i$ and $j$ in distance class $d$, $f_{xx}(d)$ is the relative frequency of occurrence for all atomic pairs in distance shell $d$, and $\sigma$ is the weight given to each observation. As discussed by Sippl[7], it was assumed that $\sigma = 0.02$.

The potential energy associated with the interaction of residues A and B denoted by $\Delta E(A, B)$ is estimated by summing the pairwise potentials between the atoms of each of these residues as follows:

$$\Delta E(A, B) = \sum_{i \in A, j \in B} \Delta E^{ij}(d) \quad (4)$$

where the sum is over all atom pairs in contact, identified via the Delaunay triangulation method.

Structural Profile of Energy (SPE): Given the 20 standard amino acids, there are 210 unique amino acid-amino acid interaction types. For each protein structure, a 210-dimensional vector was created to represent the distance-dependent energy interactions between residues. Each dimension corresponds to the energy interaction between a specific pair of amino acid types. This vector is referred to as the Structural Profile of Energy (SPE).

## Pairwise energy content from amino acid composition

While the knowledge-based potential function relies on having the three-dimensional structure of a protein, many protein structures remain undetermined experimentally. To address this, we developed a method to estimate pairwise energy content based solely on amino acid composition.

Energy Estimation: For each protein $S$ in the training set:

- $e_i^S$ denotes the energy of interactions between all residues of type $i$ and all other amino acids in protein $S$.
- The estimated energy $\widehat{e_i^S}$ is calculated using:

$$\widehat{e_i^S} = N_i^S \sum_{j=1}^{20} P_{ij} n_j^S \quad (5)$$

where, $N_i^S$ represents the frequency of amino acid type $i$ in the structure $S$, and $n_j^S = \frac{N_j^S}{L}$, is calculated as the ratio of $N_i^S$ to the total number of amino acids in $S$, denoted by $L$, and $P$ is the energy predictor matrix, delineating the dependence of amino acid $i$'s energy on the $j$ th element within the amino acid composition.

Parameter Optimization: The parameters of each row of matrix **P** were optimized by minimizing the following objective function for each amino acid type

$$Z_i = \sum_S (e_i^S - \widehat{e_i^S})^2 \quad (6)$$

By setting the partial derivatives $\frac{\partial Z_i}{\partial P_{ij}} = 0$ for all $P_{ij}$, a system of linear equations was obtained and solved using the Symbolic Math Toolbox in MATLAB.

Compositional Profile of Energy (CPE): For each pair of amino acid types $i$ and $j$, the energy $E_{ij}$ was estimated based on amino acid sequence composition:

$$E_{ij} = n_i P_{ij} n_j \quad (7)$$

where **P** is the energy predictor matrix estimated using Eq. 6. This results in a 210-dimensional vector representing energy interactions between amino acid types based on composition alone. This vector is termed the Compositional Profile of Energy (CPE) and is normalized according to protein length.

## Analysis tools and packages

All computational analyses were conducted using the versatile R programming language, with the utilization of various specialized packages tailored for specific tasks. Below is an overview of the packages and tools employed throughout our analysis:

The BIO3D software was used to read and analyze PDB files[50]. The "geometry" package was used to implement the Quickhull algorithm to find direct contacts and nearest neighbors of atoms in pdb files using the Delaunay tessellation method. Figures were generated using the ggplot2 package[51]. TM-Vec representations were generated by configuring 'tm_vec_model_cpnt' to 'tm_vec_cath_model'. The functions can be accessed at (https://github.com/tymor22/tm-vec/tree/master). TM-scores were calculated using 'tm_align' from the tmtools 0.1.1 module in python 3.10.12.

The computation times and accuracy metrics for RMSD, TM-Score, GR-align, and Yau-Hausdorff, as presented in Supplementary Table 1 and Fig. 4A, B, were sourced from Table 1 in the study by Tian et al.[28]. For this analysis, the results for CPE, SPE, and TM-Vec were generated on the same computational setup used in their study (a system with a 2.4 GHz processor and 8 GB of RAM). All other analyses were conducted on a system configured with an Intel i9, 16-core, 11th-generation processor operating at 2.6 GHz and 32 GB of RAM.

In this study, the predictions were conducted using a k-Nearest Neighbors (k-NN) classifier with k = 1, evaluated through a Leave-One-Out Cross-Validation (LOOCV) approach. In this method, each sample in the dataset was treated as a test instance, while the remaining samples constituted the training set. This comprehensive evaluation method ensures that the classifier is rigorously tested on every individual sample in the dataset. For classification, the Manhattan distance was used to determine the similarity between the test sample and all other samples, identifying the closest match (k = 1) and assigning its corresponding class label. The task involved multi-class prediction, where each protein was categorized into one of the predefined classes.

## Reporting summary

Further information on research design is available in the Nature Portfolio Reporting Summary linked to this article.

## Data availability

The data that support the findings of this study are openly available at https://github.com/mirzaie-mehdi/ProteinEnergyProfileSimilarity and on Zenodo [https://zenodo.org/records/14765519]. Source data are provided with this paper. The following is a list of datasets used in this study, along with their locations in the repository: 1 Training Set for Knowledge-Based Potential: (Train_Energy/list_with_chainID_rm_O-laps.txt). The data was obtained from http://dunbrack.fccc.edu/lab/pisces. 2 Protein Domains in ASTRAL40 (Data/csv/astral-scopedom-seqres-gd-sel-gs-bib-40-2.08.fa) and ASTRAL95 (Data/csv/astral-sco-pedom-seqres-gd-sel-gs-bib-95-2.08.fa). The data was sourced from https://scop.berkeley.edu/astral/ver=2.08. 3 The list of Bacteriocin Proteins family (Data/csv/Bacteriocin.csv). The data was obtained by requesting it from the authors of the paper[26]. 4. PDBIDs of Ferritin Superfamily (Data/csv/Ferritin_Like_seq.csv) (SCOP ID: a.25.1). 5. The list of C_terminal and Homing endonucleases (Data/csv/CT_Ho_-cathID.csv) (CATH Code: 1.10.8.10 and 3.10.28.10). 6. The list of protein domains of five superfamilies (Data/csv/fiveSF.csv) winged helix (SCOP ID: a.4.5), PH domain-like (SCOP ID: a.55.1), NTF-like (SCOP ID: d.17.4), Ubiquitin-like (SCOP ID: d.15.1), and Immunoglobulins (SCOP ID: b.1.1). 7. Covid19 spike proteins data set (Data/covidPDB/). The data was obtained from https://cov3d.ibbr.umd.edu/. 8. The list of Drug-Targets (Data/csv/41467_2019_9186_MOESM4_ESM.xlsx) including 65 anti-hypertensive drugs and their protein targets IDs downloaded from the supplementary information[22]. 9. The list of 21 mammalian hemoglobin's proteins in Globin family (Data/Globin/Globin.csv). Data was obtained from the supplementary information of the paper[23]. 10. Large-Scale SARS-CoV-2 Proteome Analysis across 28 families. The protein models can be find in the Large_Scale_SARS2 folder (Data/Large_Scale_SARS2) according to the sars_proteom column (Data/Large_Scale_SARS2/sars_proteom.csv). Data was obtained from the supplementary information of the paper[23]. Source data are provided with this paper.

## Code availability

All developed Code in this study can be found at https://github.com/mirzaie-mehdi/ProteinEnergyProfileSimilarity. and on Zenodo at https://zenodo.org/records/14765519.

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

## Acknowledgements

P.Ch. and M.M. were supported from the grants of J.O.A. - Academy of Finland (grants no. 297727, J.O.A. and 350678, J.O.A.), Sigrid Juselius Foundation, ERA-NET NEURON grant nr 352077, J.O.A., Helsinki Institute of Life Science Research Fellow, JAES Foundation grant no. 240034, J.O.A., and by HORIZON-RIA grant nr 101188432 DTRIP4H, J.O.A. In addition, J.O.A. was supported by Center of Innovative Medicine (CIMED) grant, Hjärnfonden, Team Rynkeby-God Morgon Skolloppet grant, Åhlén-stiftelsen grant, Swedish Research Council (grants no. 2019- 01578 and 2022-01093). The authors would like to thank Vilma Iivanainen, Elina Nagaeva, and Sakari Hietanen for reading the manuscript and providing valuable feedback. Open access was supported by Helsinki University Library.

## Author contributions

M.M. designed the method and supervised the research; M.M. and P.Ch. developed the mathematical formalism and analyzed data; M.M., P.Ch., and J.O.A. wrote the paper. All authors have read and agreed to the published version of the manuscript.

## Competing interests

The authors declare no competing interests.
