## [Transparent Peer Review file · Nature Communications]

A Fast approach for structural and evolutionary analysis based on energetic profile protein comparison

Corresponding Author: Dr Mehdi Mirzaie

A version of this paper was originally rejected for publication by Nature Communications, however that decision was reconsidered after appeal by the authors.

Version 1:

Reviewer comments:

Reviewer #1

(Remarks to the Author)

The authors presented a method to characterize protein-protein similarity using an energy profile. Although the proposed method is interesting, more systematic evaluations using larger datasets are needed. In addition, there are no details about how the authors obtained the energy predictor matrix to calculate sequence-based energy for each evaluation test presented in the manuscript.

Major comments:

1. More details about how the authors obtained the energy predictor matrix for calculating sequence-based energy for each evaluation test are needed. For instance, the testing dataset shouldn't be used to estimate the energy predictor matrix.
2. All the evaluations are based on small datasets. Evaluations and comparisons with other methods on larger datasets are needed.
3. Structural-based (SPE) and sequence-based (CPE) methods should be compared in each evaluation test. For example, CPE is compared with other methods in Figure 6 but not SPE. SPE is compared with other methods in Figure 7 but not CPE.
4. For the spike protein classification, the author may want to compare with a sequence similarity-based method. In addition, the order of the clusters in hierarchical clustering doesn't represent the similarity among the clusters. If the authors want to make a statement about MERS_Cov involved to SARS_Cov, then SARS_Cov2, a better way is to examine the similarity among these three groups.
5. In Figure 7, TM-Vec seems to do a better job than SPE. The authors may want to provide summary statistics to compare the performance of the two methods.

(Remarks on code availability)

Reviewer #2

(Remarks to the Author)

This is an interesting paper highlighting the use of energy profile based analysis to examine evolutionary questions. The energy profile as a way to analyze sequence structure mapping was introduced by Eisenberg in Science 253,164(1991). Soon after Wolynes and coworkers showed how such profiles could be optimized using energy landscape theory in PNAS89,9029(1992) achieving success comparable to what is discussed here, but without the evolutionary aspects. These pioneering works should be cited.

It seems also the evolutionary analysis resembles the FRUSTRAEVO work by Ferreiro and coworkers that recently appeared in Nature Communications. This work should be compared and cited.

The examples chosen are quite nice.

(Remarks on code availability)

I did not review the code.

Reviewer #3

(Remarks to the Author)

The study presents a new approach for comparing protein structures using energy profiles derived from amino acid sequences, which offer a faster and computationally efficient alternative to traditional structural alignment methods. By generating 210-dimensional profiles based on pairwise amino acid interactions, the method accurately reflects structural and evolutionary relationships among proteins, as validated by strong correlations with benchmark structural data. It effectively distinguishes protein families across hierarchical levels and has shown success in clustering proteins from viruses and bacteriocins. Additionally, the method's efficiency extends to applications in drug combination prediction, highlighting its versatility and potential in large-scale protein analysis. The manuscript is commendably clear and the logical order in which the concepts and findings are presented greatly facilitates understanding.

Comment 1: On page 5, the authors presented a useful comparison between structural and sequence-based energy calculations, suggesting that sequence-based energy estimation serves as a reliable approximation. However, it would be beneficial to address potential limitations or assumptions in this comparative analysis. In particular, discussing how sequence length and protein complexity might impact the accuracy of energy estimates would provide a more comprehensive understanding of the strengths and weaknesses of each method.

Comment 2: The manuscript effectively compares the new method with established measures (GR-Align, RMSD, TM-Score, Yau-Hausdorff distance, TM-Vec). However, additional details regarding the specific conditions or parameters used to adjust each benchmarked method, as well as any inherent biases or limitations, would enhance the reader's understanding of the comparative evaluation.

Comment 3: The reported computational efficiency of the proposed method is notable. Nevertheless, a discussion on the scalability of the method would be beneficial. Specifically, information on how computational time scales with the number of proteins or the complexity of the dataset would provide a clearer perspective on the practical applicability of the method in larger and more complex scenarios.

Comment 4: The results indicate that the energy profile-based model effectively reconstructs the phylogenetic tree. To strengthen these findings, the inclusion of statistical validation measures, such as confidence intervals or statistical significance tests, would be valuable. This addition would provide more robust support for the accuracy of the phylogenetic reconstructions.

Comment 5: The manuscript reports a discrepancy between the text and Table 1 regarding the accuracy and computation time of the 1-NN classifier based on the distance between profiles' energy (CPE). The text states an accuracy of 97% and a computation time of approximately 3 minutes, whereas Table 1 shows an accuracy of 100% and a computation time of 1 second. The authors should reconcile this discrepancy to ensure consistency and accuracy in the reporting of results.

Comment 6: The authors demonstrate that the CPE method not only matches TM-Vec in accuracy but also exhibits superior computational efficiency and interpretability. The reliance of CPE on energy profiles and pairwise amino acid interactions provides a clearer understanding of protein similarities compared to the black-box nature of deep learning techniques used in TM-Vec. Further commentary on how the CPE approach achieves this enhanced interpretability would be valuable. Specifically, elucidating how the reliance on energy profiles and amino acid interactions facilitates a more straightforward understanding of protein relationships compared to TM-Vec's deep learning model would be beneficial.

Comment 7: The methods section in general would benefit from a more comprehensive description to facilitate a clearer understanding of the methodology and enable others to replicate the study more effectively.

Comment 8: The GITHUB repository contains all the described datasets and code; however, it would benefit from additional documentation. Specifically, further explanation on how to execute and run the scripts would be valuable. Additionally, guidance on how to interpret the outputs of these scripts is necessary for users to fully understand and utilize the provided tools. The repository and the manuscript should include a comprehensive table listing all the datasets used in the manuscript. This table should detail the exact names of the datasets and provide direct pointers to the files containing them. This improvement would greatly enhance the usability and transparency of the repository.

I recommend the manuscript for publication after addressing the minor revisions, pending any final minor edits from the editorial team.

(Remarks on code availability)

The repository contains all the described datasets and code; however, it would benefit from additional documentation. Specifically, further explanation on how to execute and run the scripts would be valuable. Additionally, guidance on how to interpret the outputs of these scripts is necessary for users to fully understand and utilize the provided tools.

Furthermore, the repository should include a comprehensive table listing all the datasets used in the manuscript. This table

should detail the exact names of the datasets and provide direct pointers to the files containing them. This improvement would greatly enhance the usability and transparency of the repository.

Version 2:

Reviewer comments:

Reviewer #1

(Remarks to the Author)

The authors have addressed most of my comments and concerns. The manuscript has been significantly improved compared to the previous version.

There is one additional issue that the author may want to address. There are no details about how the authors perform the protein class prediction and how the different evaluation metrics are calculated throughout the different tasks. They only mentioned that they used kNN (k=1) to make predictions. Are these predictions performed using a leave-one-out procedure or a K-fold cross-validation procedure? Are these predictions formulated as binary or multi-class predictions? It is confusing why the class accuracy in Tables S3, S4, and S5 are exactly the same for different classes of proteins.

(Remarks on code availability)

Reviewer #3

(Remarks to the Author)

The authors have thoroughly addressed all of my revisions, and I am happy to recommend the article for publication, pending further edits by the editorial team.

(Remarks on code availability)

Comments from the Reviewers:

Reviewer 1:

The authors presented a method to characterize protein-protein similarity using an energy profile. Although the proposed method is interesting, more systematic evaluations using larger datasets are needed. In addition, there are no details about how the authors obtained the energy predictor matrix to calculate sequence-based energy for each evaluation test presented in the manuscript.

ANSWER: We thank the reviewer for appreciating our work and for very constructive feedback. We have carefully considered the comments and provide our detailed responses below. To facilitate the review, we have included new data and figures/tables into our point-by-point responses below and updated the revised manuscript accordingly. The new texts are indicated in red both below and in the revised manuscript.

1) More details about how the authors obtained the energy predictor matrix for calculating sequence-based energy for each evaluation test are needed. For instance, the testing dataset shouldn't be used to estimate the energy predictor matrix.

ANSWER: We thank the reviewer for this suggestion and have now excluded the 129 proteins that overlapped with the test sets from the training set. The knowledge-based potential function and energy predictor matrix are now recalculated using only the remaining training set proteins, ensuring that the test data is not involved in the matrix estimation process. In the revised manuscript, we have also updated the Methods section (Page 17-19) and added new **Fig. 1**, which illustrates the steps for estimating the predictor matrix and energy profile.

The recalculated predictor matrix showed a correlation of 0.99990 with the original one, indicating that the exclusion of overlapping proteins had a negligible impact on the energy predictions. In addition, we regenerated all figures and reran the method across all test sets, with results remaining consistent with our previous findings, confirming the robustness of our approach. The new text is indicated in red both below and in the revised manuscript.

Dataset Preparation

A curated dataset of non-redundant protein chains was generated using PISCES⁴⁷ from the Protein Data Bank (PDB). The dataset was selected based on the following criteria:

- **Pairwise sequence identity:** Less than 50% to ensure non-redundancy.
- **Resolution:** Higher than 1.6 Å to guarantee structural accuracy.
- **R-factor:** Below 0.25 to ensure reliable crystallographic data.
- **Protein length:** Between 40 and 1,000 residues to include proteins of varying sizes while excluding excessively short or long chains.
- **Overlap:** Proteins overlapping with the test sets from this manuscript were removed from the training set.

These filtered proteins were utilized to train and calculate the knowledge-based potential function as follows.

Pairwise Distance-Dependent Knowledge-Based Potential

Knowledge-based potentials are derived from databases of known protein structures and are essential for estimating the energies of pairwise interactions. These potentials can be based on various factors, including distance dependencies, dihedral angles, and accessible surface areas⁸. In this study, we employed a distance-dependent potential function where atomic contacts were identified using the tessellation method^{9, 48, 49} as follows:

Contact Identification:

1. **Representation:** All amino acids in each protein chain were represented by their heavy atoms (excluding hydrogen atoms).
2. **Delaunay Tessellation:** A Delaunay tessellation of the resulting point set was computed using Qhull⁵⁰, identifying neighboring atoms based on spatial proximity.
3. **Defining Contacts:** Two atoms were considered to be in contact if they are connected by an edge in the Delaunay triangulation. This implies that they are not shielded from each other by other atoms, ensuring direct interaction without obstruction.
4. **Distance Shells:** The distances between contacting atoms were divided into 30 discrete shells, starting at 0.75 Å with each shell having a width of 0.5 Å. This binning allows for the extraction of distance-dependent interaction potentials.

Atom Types: A total of 167 atom types were considered by treating non-hydrogen atoms as distinct based on their specific amino acid residues.

Energy Calculation: The potential energy between two atoms i and j at distance d was calculated using the following equation⁷:

$$\Delta E^{ij}(d) = RT \left[\ln(1 + M_{ij}\sigma) - \ln(1 + M_{ij}\sigma \left(\frac{f_{ij}(d)}{f_{xx}(d)}\right)) \right] \quad (1)$$

where RT is constant and equal to 0.582 kcal/mole. M_{ij} is the number of observations for atomic pair i and j , $f_{ij}(d)$ is the relative frequency of occurrence for i and j in distance class d , $f_{xx}(d)$ is the relative frequency of occurrence for all atomic pairs in distance shell d , and σ is the weight given to each observation. As discussed by Sippl⁷, it was assumed that $\sigma = 0.02$.

The potential energy associated with the interaction of residues A and B denoted by $\Delta E(A, B)$ is estimated by summing the pairwise potentials between the atoms of each of these residues as follows:

$$\Delta E(A, B) = \sum_{i \in A, j \in B} \Delta E^{ij}(d) \quad (2)$$

where the sum is over all atom pairs in contact, identified via the Delaunay triangulation method.

Structural Profile of Energy (SPE): Given the 20 standard amino acids, there are 210 unique amino acid-amino acid interaction types. For each protein structure, a 210-dimensional vector was created to represent the distance-dependent energy interactions between residues. Each dimension corresponds to the energy interaction between a specific pair of amino acid types. This vector is referred to as the **Structural Profile of Energy (SPE)**.

Pairwise Energy Content from Amino Acid Composition

While the knowledge-based potential function relies on having the three-dimensional structure of a protein, many protein structures remain undetermined experimentally. To address this, we developed a method to estimate pairwise energy content based solely on amino acid composition.

Energy Estimation: For each protein S in the training set:

- e_i^S denotes the energy of interactions between all residues of type i and all other amino acids in protein S .
- The estimated energy \widehat{e}_i^S is calculated using:

$$\widehat{e}_i^S = N_i^S \sum_{j=1}^{20} P_{ij} n_j^S \quad (3)$$

where, N_i^S represents the frequency of amino acid type i in the structure S , and $n_j^S = \frac{N_j^S}{L}$, is calculated as the ratio of N_j^S to the total number of amino acids in S , denoted by L , and P is the energy predictor matrix, delineating the dependence of amino acid i 's energy on the j th element within the amino acid composition.

Parameter Optimization: The parameters of each row of matrix P were optimized by minimizing the following objective function for each amino acid type

$$Z_i = \sum_S (e_i^S - \widehat{e}_i^S)^2 \quad (4)$$

By setting the partial derivatives $\frac{\partial Z_i}{\partial P_{ij}} = 0$ for all P_{ij} , a system of linear equations was obtained and solved using the Symbolic Math Toolbox in MATLAB.

Compositional Profile of Energy (CPE): For each pair of amino acid types i and j , the energy E_{ij} was estimated based on amino acid sequence composition:

$$E_{ij} = n_i P_{ij} n_j \quad (5)$$

where P is the energy predictor matrix estimated using equation 4. This results in a 210-dimensional vector representing energy interactions between amino acid types based on composition alone. This vector is termed the **Compositional Profile of Energy (CPE)** and is normalized according to protein length.

Fig. 1 | Development of knowledge-based potential function and profile of energy. A) Construction of the knowledge-based potential function. B) Estimation of the predictor matrix P. C) Construction of the structural profile of energy (SPE) based on protein structure. D) Construction of the compositional profile of energy (CPE) based on protein sequence.

At the beginning of the results section on page 5, we provided an explanation of Fig. 1 as follows:

Knowledge-based potentials are derived from databases of known protein structures. Various potential functions, such as distance-dependent, dihedral angles, and accessible surface energies leverage information from known protein structures to estimate energies of pairwise interactions^{7, 8}. In this study, a knowledge-based potential function was developed using a curated dataset of non-redundant protein chains from the Protein Data Bank (PDB), selected for high structural accuracy and diversity as detailed in the Method section. Pairwise distance-dependent potentials were calculated based on atomic interactions, identified through Delaunay tessellation, with energies derived from the frequency of atomic contacts at various distance intervals. The energy between atom pairs was computed following equation (1) in the Method section (Fig. 1A). Furthermore, an energy predictor matrix was created to estimate the pairwise energy content solely from amino acid composition (Fig. 1B). Given the 20 amino acids in proteins, equation (2) was applied to represent each protein structure using 210 distinct pairwise interaction types (Fig. 1C),

leading to the generation of the 210-dimensional Structural Profile of Energy (SPE). Additionally, equation (5) was employed to compute the Compositional Profile of Energy (CPE) based on protein sequences (Fig. 1D). For each pair of proteins, the Manhattan distance between the profiles of energies is considered a measure of dissimilarity between them.

2) All the evaluations are based on small datasets. Evaluations and comparisons with other methods on larger datasets are needed.

ANSWER: We have now conducted an additional evaluation using a large dataset and compared our results with the TM-Vec method. Given that structure-based methods such as RMSD and TM-score are time-intensive, we focused on TM-Vec for its balance of efficiency and accuracy. For the larger dataset, we employed the coronavirus dataset previously generated and analyzed by Freiberger et al., as suggested by Reviewer 2. These new results are now included in **Section 3.8 (Page 13-14)** of the manuscript as displayed below:

3.8 Large-Scale Application of Family Detection in Coronaviruses

To evaluate our method on a larger dataset, we utilized a coronavirus dataset that was previously generated and analyzed by Freiberger et al.²³ Initially, they retrieved homologous sequences from the SARS-CoV-2 reference genome MN985325, with low-quality and non-Coronaviridae sequences excluded⁴³. Then sequences were aligned using MAFFT software⁴⁴, with non-structural protein sequences trimmed to focus on regions specifically relevant to SARS-CoV-2. These aligned sequences were then clustered using CD-Hit⁴⁵ to ensure high similarity within each cluster. Structural models for these sequences were generated using AlphaFold2, retaining only those that met stringent quality criteria. The high-quality models were subsequently grouped into subfamilies using S3Det software⁴⁶, allowing for the identification of Specificity Determining Positions (SDPs) within the proteins. This meticulous process resulted in a final set of 28 high-quality protein families, comprising 4,405 protein models. For each protein in this dataset, we computed the energy profiles CPE and SPE, along with the distance between these profiles and the cosine similarity between pairs of TM-Vec representations. The 1-nearest neighbor (1-NN) method was then utilized to classify the proteins into different families. The results, shown in Table 5, include metrics for accuracy and F1-score, demonstrating the effectiveness of our model. Both methods achieved performance levels near 100%, with CPE offering faster performance. Detailed outcomes of the 1-NN classification are provided in Supplementary Tables S3-S5, and the UMAP projections of SPE, CPE, and TM-Vec representations are displayed in Supplementary Figures S6-S8.

The Papain-like Protease (PLPro) domain plays a crucial role in viral replication by catalyzing the proteolysis of viral polyproteins. In addition, PLPro interacts with two host proteins, ubiquitin (Ub) and the ubiquitin-like interferon-stimulated gene 15 protein (ISG15), allowing the virus to evade or weaken the host immune response. In this dataset, PLPro was divided into four subfamilies aligned with the Betacoronavirus subgenera: Sarbecovirus (n = 31), Nobecovirus (n = 11), Merbecovirus (n = 35), and Embecovirus (n =

45)²³. The UMAP representation for both CPE and SPE is shown in Fig. 13, where proteins from each subfamily are clearly clustered together.

Table 5 | Large Scale Analysis of the SARS-CoV-2 Proteome across 28 families, encompassing 4,405 proteins. The overall accuracy, F1 score, and computation time for detecting families using the 1-NN classifier.

Method	Time (Sec)	Accuracy	F1 Measure
CPE	49	0.9968	0.9946
SPE	2715	0.9973	0.9903
TM_Vec	685	0.9966	0.9925

Fig. S6 | UMAP Visualization of Energy Profiles in Large-Scale SARS-Cov2 data set. The UMAP projection of Structural Energy Profiles (SPE) on 28 protein families with a total of 4,405 protein models. UMAP plot was generated using parameters `n_neighbors = 150` and `min_dist = 0.1`.

Fig. S7 | UMAP Visualization of Energy Profiles in Large-Scale SARS-Cov2 data set. The UMAP projection of Compositional Energy Profiles (CPE) on 28 protein families with a total of 4,405 protein models. UMAP plot was generated using parameters `n_neighbors = 150` and `min_dist = 0.1`.

Fig. S8 | UMAP Visualization of Energy Profiles in Large-Scale SARS-Cov2 data set. The UMAP projection of TM-Vec on 28 protein families with a total of 4,405 protein models. UMAP plot was generated using parameters `n_neighbors = 150` and `min_dist = 0.1`.

Fig. 13 | Papain-like Protease (PLPro) domains across Betacoronavirus subgenera. The UMAP projection shows clustering of PLPro domains from Sarbecovirus (n = 31), Nobecovirus (n = 11), Merbecovirus (n = 35), and Embecovirus (n = 45) using SPE, CPE, and TMVec representations. n_neighbors = 13, min_dist = 0.5.

3) Structural-based (SPE) and sequence-based (CPE) methods should be compared in each evaluation test. For example, CPE is compared with other methods in Figure 6 but not SPE. SPE is compared with other methods in Figure 7 but not CPE.

ANSWER: We have now included comparisons between **CPE** and **SPE** across the appropriate evaluation tests. Results are visualized on new **Fig. 7** (previously **Fig. 6**) and in **Table 1**. We find that (Table 1 and new Fig. 7) CPE method achieves a computation time faster than TM-Vec and that CPE and SPE methods significantly outperform GR-Align, RMSD, TM-Score, and YH in terms of accuracy and efficiency, highlighting advantages of CPE and SPE. The following discussion has been added to the manuscript on page 8.

As shown in Table 1 and Fig. 7, both CPE and TM-Vec achieved 100% accuracy in distinguishing between the two protein families. However, the CPE method significantly reduced computation time, completing the task in just one second compared to TM-Vec's 67 seconds. Furthermore, CPE, SPE, and TM-Vec outperformed GR-Align, RMSD, TM-score, and YH in terms of accuracy. Notably, our approach eliminates the need for structural alignment by relying on energy profile calculations and distance measurements.

Table 1 | The accuracy and computation time for 1-NN classifier based on GR-Align, RMSD, TM-score, Yau-Hausdorff distance, TM-Vec, and the distance between profiles of energy SPE and CPE as a measure of protein dissimilarity.

Method	Accuracy	Time
GR-Align	62.3%	2 min
RMSD	59.2%	1 h
TM-Score	61.5%	9 h 20 min

YH (10 Rotation)	70.8%	10 min
YH (2500 Rotation)	81.5%	4h 10 min
TM-Vec	100%	67 sec
CPE	100%	1 sec
SPE	99%	187 sec

Fig. 7 | Performance and Computational Efficiency of Protein Dissimilarity Measures. A) Time versus accuracy for the 1-NN classifier using GR-Align, RMSD, TM-score, Yau-Hausdorff distance, TM-Vec, and the distance between energy profiles SPE and CPE as measures of protein dissimilarity. B) Running times of the evaluated methods, scaled to 12 hours, with an inset zooming in on the region indicated by the dashed circle. The entire circle represents 80 seconds. Each method is represented by different colors as indicated in the figure legend.

We have also updated Table 2 to include the SPE method results for distinguishing between five distinct SCOP superfamilies: Winged Helix (a.4.5), PH Domain-like (b.55.1), NTF-like (d.17.4),

Ubiquitin-like (d.15.1), and Immunoglobulins (b.1.1). The following discussion has been added to the manuscript on page 8.

To evaluate the effectiveness of energy profiles in protein superfamily classification, we examined five distinct SCOP superfamilies: winged helix (a.4.5), PH domain-like (b.55.1), NTF-like (d.17.4), Ubiquitin-like (d.15.1), and Immunoglobulins (b.1.1)²⁹. Our classification approach utilized both energetic profiles and TM-Vec representations as features, applying 1-nearest neighbor (1-NN) classifier. The results, summarized in Table 2, report metrics for accuracy and F1-score, demonstrating the model's effectiveness. All methods achieved performance levels approaching 100%, as shown in Table 2. The CPE method, in particular, not only matched TM-Vec in terms of accuracy but also exhibited faster computation times.

Table 2 | Total accuracy and F1 measure for each of the five superfamilies by 1-NN based on CPE, SPE, and TM-Vec.

Method	Time (Sec)	Accuracy	F1 Measure				
			wigend_helix	PH.domain-like	NTF-like	Ubiquitin-like	Immunoglobulins
CPE	103	0.98	0.98	0.96	0.99	0.99	0.99
SPE	3524	0.98	0.98	0.96	0.99	0.98	0.98
TM_Vec	955	0.99	0.99	1	1	0.99	0.99

We have also included the CPE results for the phylogenetic network reconstruction of the ferritin-like superfamily in Fig. 9 (previously Fig. 7). A more detailed comparison of the performance of these methods is provided in the response to the comment 5.

4) For the spike protein classification, the author may want to compare with a sequence similarity-based method. In addition, the order of the clusters in hierarchical clustering doesn't represent the similarity among the clusters. If the authors want to make a statement about MERS_Cov involved to SARS_Cov, then SARS_Cov2, a better way is to examine the similarity among these three groups.

ANSWER: Thank you for this very helpful comment. We have revised Section 3.5 to address the requirements, and Figure 10 has been updated accordingly. In response, we included a phylogenetic tree constructed using sequence similarity-based methods and added the following discussion to the manuscript on page 11.

Initially, we conducted a multiple sequence alignment of spike proteins using the ClustalW method via the msa package³⁵ in R. Protein distances were calculated with the seqinr package³⁶, using sequence identity as the distance metric. Phylogenetic trees were subsequently generated using the UPGMA method from the phangorn package³³, and the resulting tree based on sequence similarity is presented in Fig. 10A. To further examine structural variations and relationships among the spike glycoproteins, we applied structural methods including RMSD, TM-Score, and SPE, alongside sequence-based methods such as CPE and TM-Vec, to calculate pairwise distances and cluster the spike glycoproteins into three distinct groups corresponding to SARS-CoV, MERS-CoV, and SARS-CoV-2. This clustering offers a visual representation of the structural and evolutionary relationships within this protein family, as depicted in Fig. 10A-F.

All methods consistently showed a clear separation between the SARS-CoV/SARS-CoV-2 lineage and MERS-CoV, which belongs to a different coronavirus subgenus. However, as illustrated in Fig. 10C, E, and F, methods like RMSD, TM-Score, and TM-Vec were less effective in capturing these evolutionary patterns. For instance, certain SARS-CoV proteins were misclassified and appeared distant from their respective groups (highlighted by orange circles in Fig. 10E). Similarly, the TM-Vec method misclassified some SARS-CoV proteins and failed to accurately group MERS-CoV proteins (Fig. 10C).

Fig. 10 | Clustering analysis of spike glycoprotein structures from SARS-CoV, SARS-CoV-2, and MERS-CoV. The dendrograms depict the clustering of spike glycoprotein structures from the three viruses: SARS-CoV, SARS-CoV-2, and MERS-CoV. The clustering is based on pairwise distances calculated from different methods: **A)** protein sequence, **B)** CPE, **C)** TM-Vec, **D)** SPE, **E)** RMSD, and **F)** TM-Score. The leaves of each tree are color-coded to indicate the originating virus for each spike glycoprotein structure. **G)** Displays the ARI values for each method, **H)** shows the running time associated with each method scaled to 12 hours, with an inset zooming in on the region indicated by the dashed circle. The entire circle represents 80 seconds, and **I)** presents the average distance between the three virus groups as calculated by each method.

We also computed the average distance between the three virus groups—SARS-CoV, SARS-CoV-2, and MERS-CoV—across all six methods. All methods consistently show that SARS-CoV-2 is more closely related to SARS-CoV than to MERS-CoV (Fig. 10I). Additionally, the distance between SARS-CoV-2 and MERS-CoV is nearly identical to the distance between SARS-CoV and MERS-CoV.

5) In Figure 7, TM-Vec seems to do a better job than SPE. The authors may want to provide summary statistics to compare the performance of the two methods.

ANSWER: Thank you for your valuable comment. In line with the suggestion from **comment 3**, we have now included the results for the **CPE** method in the comparison.

Fig. 9 | Phylogenetic network reconstruction of the ferritin-like superfamily. A) Schematic representation of the relationships among major ferritin-like protein families. B) Phylogenetic network reconstructed using SPE. C) Neighbor-joining tree generated based on the average distances between subgroups using SPE. D) Phylogenetic network reconstructed using TM-Vec. E) Neighbor-joining tree generated using average distances between subgroups with TM-Vec. F) Phylogenetic network reconstructed using CPE. G) Neighbor-joining tree based on the average distances between subgroups

using CPE. The red dotted line highlights the separation between two SCOP families: ferritins (SCOP ID a.25.1.1), which includes the Bacterioferritin, Ferritins, Dps, and Rubrerythrin subgroups, and the Ribonucleotide Reductase-like family (SCOP ID a.25.1.2), which includes the BMM-alpha, BMM-beta, Fatty_acid, and RNRR2 subgroups. The inferred arrangement of subfamilies using each method is shown to the right of C, E, and G.

To provide a comprehensive evaluation, we computed the average distances between proteins from each of the eight subfamily groups using three methods: CPE, SPE, and TM-Vec. These distance matrices were used to construct neighbor-joining phylogenetic trees, as shown in new **Fig. 9 C, E, and G**. To assess the accuracy of the subfamily arrangements, we compared the inferred phylogenetic trees with the manually annotated subfamily relationships, with color-coding corresponding to subgroups in **Fig. 9A**. The order of subgroups in the manually annotated network was then compared with the order produced by SPE, TM-Vec, and CPE (**Fig. 9 C, E, G**). We found that while TM-Vec performs well in clustering similar proteins, it struggles to accurately reflect the evolutionary relationships between subfamilies. For instance, in the a.25.1.2 family, TM-Vec incorrectly places the subgroup BMM-b first, whereas it should appear last in the manual network. Similarly, in the a.25.1.1 family, although TM-Vec correctly identifies the initial branch, it misorders the remaining subgroups. To quantify these deviations, we calculated the Spearman correlation between the predicted branching orders from each method and the manual network. These correlations, summarized in Table 3, demonstrate that both SPE and CPE outperform TM-Vec in aligning with the manually annotated network. Notably, SPE achieves a perfect correlation in the a.25.1.1 family, while TM-Vec shows a negative correlation in the a.25.1.2 family, illustrating its limitations.

The following discussion has been added to the manuscript at page 10:

Using the same protein structures from this superfamily as employed by Malik et al.³⁰ and Lelievre et al.³¹, we reconstructed the phylogeny of these proteins based on energy profiles (CPE and SPE) and the TM-Vec method. The dataset focuses on the SCOP Ferritin-like superfamily (a.25.1), which includes two curated families: Ferritin (a.25.1.1), containing ferritins, bacterioferritins, and Dps proteins, and Ribonucleotide Reductase-like [RNR] (a.25.1.2), comprising the RNR R2 subunit, BMM, and fatty acid desaturases. Phylogenetic trees constructed from SPE, TM-Vec, and CPE data using the phangorn package³³ and visualized with SplitTree software³⁴ are shown in Figures 9B, 9D, and 9F.

In line with the reference in Figure 9A, a notable observation is that all three phylogenetic trees exhibit two main branches (marked by dashed lines), representing the two families a.25.1.1 and a.25.1.2. This outcome highlights the effectiveness of the methods in accurately differentiating the superfamily into its respective families. Delving into specifics, the family a.25.1.1 (depicted by orange color triangles) further divides into four subgroups: "ferritins", "Dps", "Rubrerythrin", and "Bacterioferritins" indicated by distinct colors in Fig. 9A. On the other hand, the second branch related to the a.25.1.2 family (dark blue triangles), despite SCOP and CATH assigning these proteins to a unified RNR-like family, reveals three distinct families according to Pfam—Phenol_Hydrox (PF02332), Ribonuc_red_sm (PF00268), and Fatty acid desaturase (PF03405). The protein groupings presented by Lelievre et al.³¹ in Fig. 9A are color-coded, corresponding to the colors used in Fig. 9B- G.

For a comprehensive comparison with the manual annotation in Figure 9A, we calculated the average inter-protein distances across the eight subfamily groups using three distinct methods: CPE, SPE, and TM-Vec. These distance matrices were used to construct neighbor-joining phylogenetic trees, as shown in Fig. 9 C, E, and G. To assess the accuracy of the subfamily arrangements, we compared the inferred phylogenetic trees with the manually annotated subfamily relationships, with color-coding corresponding to subgroups in Fig. 9A. The order of subgroups in the manually annotated network was then compared with the order produced by SPE, TM-Vec, and CPE (Fig. 9 C, E, G). We found that while TM-Vec performs well in clustering similar proteins, it struggles to accurately reflect the evolutionary relationships between subfamilies. For instance, in the a.25.1.2 family, TM-Vec incorrectly places the subgroup BMM-b first, whereas it should appear last in the manual network. Similarly, in the a.25.1.1 family, although TM-Vec correctly identifies the initial branch, it misorders the remaining subgroups. To quantify these deviations, we calculated the Spearman correlation between the predicted branching orders from each method and the manual network. These correlations, summarized in Table 3, demonstrate that both SPE and CPE outperform TM-Vec in aligning with the manually annotated network. Notably, SPE achieves a perfect correlation in the a.25.1.1 family, while TM-Vec shows a negative correlation in the a.25.1.2 family, illustrating its limitations. Our findings indicate that the energy-based phylogenies within the ferritin-like superfamily reveal substantial relationships among its members, aligning closely with established evolutionary connections and functional roles suggested by Malik et al.³⁰, and Lelievre et al.³¹ showed in Fig. 9A.

Table 3 | Spearman Correlation Between the Manual Network and Predicted Branching Orders by SPE, TM-Vec, and CPE.

Method	a.25.1.1	a.25.1.2
CPE	0.8	0.6
SPE	1	0.6
TM_Vec	0.2	-0.4

These updates provide a more comprehensive comparison between the sequence-based and structure-based approaches, as requested, offering a clearer understanding of their relative performance across all tests.

Reviewer 2:

This is an interesting paper highlighting the use of energy profile-based analysis to examine evolutionary questions. The energy profile as a way to analyze sequence structure mapping was introduced by Eisenberg in *Science* 253,164(1991). Soon after Wolynes and coworkers showed how such profiles could be optimized using energy landscape theory in *PNAS*89,9029(1992) achieving success comparable to what is discussed here, but without the evolutionary aspects. These pioneering works should be cited.

It seems also the evolutionary analysis resembles the FRUSTRAEVO work by Ferreiro and coworkers that recently appeared in *Nature Communications*. This work should be compared and cited. The examples chosen are quite nice.

ANSWER: We would like to thank the reviewer for thoughtful and constructive feedback. We agree that the pioneering studies by Eisenberg (1991) and Wolynes et al. (1992), should be cited and contextualized properly and have done so in the revised version. To facilitate the review, we have included new data and figures/tables into our point-by-point responses below and updated the revised manuscript accordingly. The new texts are indicated in red both below and in the revised manuscript.

We have now incorporated the following discussion into the introduction (Page 3-4 of the revised ms):

The concept of using energy profiles to evaluate protein structures was initially introduced by Eisenberg¹⁴, who developed a method for mapping amino acid sequences to structural folds based on energy profiles. This approach enabled an early computational framework for assessing the compatibility of protein sequences with specific structural conformations. Soon after, Wolynes and co-workers¹⁵ expanded this study by applying energy landscape theory, utilizing optimized Hamiltonians to predict protein folding pathways. They introduced the spin-glass model to navigate the complex energy landscape of protein folding, ensuring that the native fold represents a global energy minimum. These pioneering methods laid the groundwork for modern approaches that predict protein structures using energy-based techniques.

Regarding your second point about the evolutionary analysis resembling the FRUSTRAEVO work by Ferreiro and coworkers, we acknowledge the relevance of this recent study.

Ferreiro et al. developed a method to assess frustration levels across multiple sequence alignments (MSAs) using various frustration indices, such as the Single Residue Frustration Index and pairwise interaction indices, to identify conserved frustration patterns. This approach is implemented through tools like FrustraEvo and Frustratometer, which provide detailed analyses of evolutionary divergence within protein families. They demonstrated how local frustration can influence protein foldability and function, with case studies on α and β globins, as well as the SARS-CoV-2 dataset. By assigning a frustration score to each position, their method allowed for a comprehensive analysis of the SARS-CoV-2 proteome, identifying functionally relevant sites within viral proteins based on conserved frustration patterns. In contrast, our study assigns an energy profile to each protein rather than generating a score for individual positions. Our method represents each protein as a comprehensive energy profile, asserting that proteins within different families, superfamilies, or folds exhibit different energy profiles. While both approaches share the assumption that local energetic frustration reflects evolutionary and functional constraints, the key difference lies in implementation: Ferreiro et al.'s method assigns frustration scores to each position, enabling the identification of critical residues for protein function. In our approach,

however, we analyze the overall energy profile of proteins, focusing on broader structural and evolutionary patterns rather than pinpointing individual functional residues.

We utilized a coronavirus dataset generated and analyzed by Freiberger et al. In the revised manuscript, we have included a new section (Section 3.8, **Page 13-14**) titled "Large-Scale Application of Family Detection in Coronaviruses," which presents the results of our method in comparison with TM-Vec. The section is as follows:

3.8 Large-Scale Application of Family Detection in Coronaviruses

To evaluate our method on a larger dataset, we utilized a coronavirus dataset that was previously generated and analyzed by Freiberger et al.²³ Initially, they retrieved homologous sequences from the SARS-CoV-2 reference genome MN985325, with low-quality and non-Coronaviridae sequences excluded⁴³. Then sequences were aligned using MAFFT software⁴⁴, with non-structural protein sequences trimmed to focus on regions specifically relevant to SARS-CoV-2. These aligned sequences were then clustered using CD-Hit⁴⁵ to ensure high similarity within each cluster. Structural models for these sequences were generated using AlphaFold2, retaining only those that met stringent quality criteria. The high-quality models were subsequently grouped into subfamilies using S3Det software⁴⁶, allowing for the identification of Specificity Determining Positions (SDPs) within the proteins. This meticulous process resulted in a final set of 28 high-quality protein families, comprising 4,405 protein models. For each protein in this dataset, we computed the energy profiles CPE and SPE, along with the distance between these profiles and the cosine similarity between pairs of TM-Vec representations. The 1-nearest neighbor (1-NN) method was then utilized to classify the proteins into different families. The results, shown in Table 5, include metrics for accuracy and F1-score, demonstrating the effectiveness of our model. Both methods achieved performance levels near 100%, with CPE offering faster performance. Detailed outcomes of the 1-NN classification are provided in Supplementary Tables S3-S5, and the UMAP projections of SPE, CPE, and TM-Vec representations are displayed in Supplementary Figures S6-S8.

The Papain-like Protease (PLPro) domain plays a crucial role in viral replication by catalyzing the proteolysis of viral polyproteins. In addition, PLPro interacts with two host proteins, ubiquitin (Ub) and the ubiquitin-like interferon-stimulated gene 15 protein (ISG15), allowing the virus to evade or weaken the host immune response. In this dataset, PLPro was divided into four subfamilies aligned with the Betacoronavirus subgenera: Sarbecovirus (n = 31), Nobecovirus (n = 11), Merbecovirus (n = 35), and Embecovirus (n = 45)²³. The UMAP representation for both CPE and SPE is shown in Fig. 13, where proteins from each subfamily are clearly clustered together.

Table 5 | Large Scale: Analysis of the SARS-CoV-2 Proteome across 28 families, encompassing 4,405 proteins. The overall accuracy, F1 score, and computation time for detecting families using the 1-NN classifier.

Method	Time (Sec)	Accuracy	F1 Measure
CPE	49	0.9968	0.9946

SPE	2715	0.9973	0.9903
TM_Vec	685	0.9966	0.9925

Fig. S6 | UMAP Visualization of Energy Profiles in Large-Scale SARS-Cov2 data set. The UMAP projection of Structural Energy Profiles (SPE) on 28 protein families with a total of 4,405 protein models. UMAP plot was generated using parameters $n_neighbors = 150$ and $min_dist = 0.1$.

Fig. S7 | UMAP Visualization of Energy Profiles in Large-Scale SARS-Cov2 data set. The UMAP projection of Compositional Energy Profiles (CPE) on 28 protein families with a total of 4,405 protein models. UMAP plot was generated using parameters $n_neighbors = 150$ and $min_dist = 0.1$.

Fig. S8 | UMAP Visualization of Energy Profiles in Large-Scale SARS-Cov2 data set. The UMAP projection of TM-Vec on 28 protein families with a total of 4,405 protein models. UMAP plot was generated using parameters `n_neighbors = 150` and `min_dist = 0.1`.

Fig. 13 | Papain-like Protease (PLPro) domains across Betacoronavirus subgenera. The UMAP projection shows clustering of PLPro domains from Sarbecovirus ($n = 31$), Nobecovirus ($n = 11$), Merbecovirus ($n = 35$), and Embecovirus ($n = 45$) using SPE, CPE, and TMVec representations. $n_neighbors = 13$, $min_dist = 0.5$.

We also used a dataset of the two subfamilies, α and β globins, generated and analyzed by Freiberger et al., with the results included on page 9 of the manuscript.

To further validate our approach, we also analyzed the two subfamilies, α and β globins, which are part of the hemoglobin biological unit. Although α and β globins are closely related and share a common evolutionary origin, they have distinct and well-characterized functions within the hemoglobin $\alpha\beta_2$ tetramer, despite their highly similar structures. Freiberger et al.²³ utilized a non-redundant set of experimental structures representing 21 mammalian hemoglobins (both α - and β -globins), and their analysis revealed distinct patterns of conserved energetic frustration, corresponding to their divergent functional roles in hemoglobin. Notably, specific residues exhibited highly frustrated interactions in one family while maintaining stability in the other, indicating evolutionary adaptations that reflect the unique structural and functional demands of each globin subunit. We computed the CPE, SPE, and TM-Vec representations for this dataset. Fig. 8 presents the UMAP projection for these proteins, where all methods effectively differentiate between the α and β globins.

Fig. 8 | The UMAP projection of α and β globins from the hemoglobin biological unit. The figure shows the clustering of 21 mammalian hemoglobins, divided into α -globins and β -globins, using CPE, SPE, and TM-Vec representations. $n_neighbors = 13$, $min_dist = 0.1$

Reviewer 3:

The study presents a new approach for comparing protein structures using energy profiles derived from amino acid sequences, which offer a faster and computationally efficient alternative to traditional structural alignment methods. By generating 210-dimensional profiles based on pairwise amino acid interactions, the method accurately reflects structural and evolutionary relationships among proteins, as validated by strong correlations with benchmark structural data. It effectively distinguishes protein families across hierarchical levels and has shown success in clustering proteins from viruses and bacteriocins. Additionally, the method's efficiency extends to applications in drug combination prediction, highlighting its versatility and potential in large-scale protein analysis. The manuscript is commendable clear and the logical order in which the concepts and findings are presented greatly facilitates understanding.

ANSWER: We would like to thank the reviewer for the positive feedback and for recognizing the value of our work. To facilitate the review, we have included new data and figures/tables into our point-by-point responses below and updated the revised manuscript accordingly. The new texts are indicated in red both below and in the revised manuscript.

1) On page 5, the authors presented a useful comparison between structural and sequence-based energy calculations, suggesting that sequence-based energy estimation serves as a reliable approximation. However, it would be beneficial to address potential limitations or assumptions in this comparative analysis. In particular, discussing how sequence length and protein complexity might impact the accuracy of energy estimates would provide a more comprehensive understanding of the strengths and weaknesses of each method.

ANSWER: We thank the reviewer for the valuable suggestion to address the limitations and assumptions in our comparison between structural and sequence-based energy calculations. We have added the following section and discussion to the revised version of the manuscript at page 5-6.

Fig. 2A and 2B depict the strong correlation between total energy derived from structural data (y-axis) and energy estimated from sequence data (x-axis) for the ASTRAL40 and ASTRAL95 datasets. The high correlation coefficient observed suggests that sequence-based energy estimation provides a reliable approximation, which can be effectively applied in cases where the protein structure is unknown. Furthermore, we calculated the total energy for both protein sequences and their corresponding structures using protein domains from the ASTRAL40 dataset and analyzed the differences between these estimates. As shown in Fig. 2C, we specifically examined the correlation between energy differences and protein length. The results indicate no significant correlation, demonstrating that the accuracy of sequence-based total energy estimates is independent of protein length. This confirms that sequence-based energy calculations can serve as a robust approximation of structural energies across proteins of varying lengths. For each pair of domains within the ASTRAL40 and ASTRAL95 datasets, the distances between their energy profiles were calculated using both structural and sequence-based energy estimates. In Fig. 2D and 2E, the x-axis represents the distance between Compositional Profiles of Energies (CPE), while the y-axis represents the distance between Structural Profiles of Energies (SPE). The strong correlation observed

between the two approaches indicates that sequence-based energy estimation is sufficiently reliable. To further support this conclusion, we extended our analysis to investigate energy discrepancies across all interaction types. For each interaction type, we calculated the differences between energy estimates derived from sequence and structure. As shown in Fig. 2F, 96% of the interaction types displayed a correlation of less than 0.5 between energy differences and protein length, indicating that protein length does not significantly affect the accuracy of energy estimates for most interaction types. This reinforces our conclusion that sequence-based energy approximations are robust across diverse protein interactions. Supplementary Fig. S1 provides scatter plots for all 210 interaction types. However, while protein length does not appear to influence accuracy, we acknowledge that protein complexity—such as folding patterns, structural heterogeneity, and conformational dynamics—may indeed play a role in the precision of energy estimates. Therefore, exploring the impact of protein complexity on energy estimations will be a valuable direction for future research.

Fig. 2 | Sequence-Structure relationship. The correlation between total energy estimates derived from protein structure and sequence for protein domains within A) ASTRAL40 and B) ASTRAL95 data sets. C) The correlation between the difference in total energy (from sequence and structure) and protein length. The correlation between the distances of profile of energy estimated from sequence (CPE) and structure (SPE) for all pairs of domains in D) ASTRAL40 and E) ASTRAL95. F) Histogram showing the distribution of correlation coefficients between the difference in energy estimates (from sequence and structure) and protein length across all 210 pairwise interactions.

2) The manuscript effectively compares the new method with established measures (GR-Align, RMSD, TM-Score, Yau-Hausdorff distance, TM-Vec). However, additional details regarding the specific conditions or parameters used to adjust each benchmarked method, as well as any inherent biases or limitations, would enhance the reader's understanding of the comparative evaluation.

ANSWER: We would like to thank the reviewer for very helpful suggestions. We have revised the manuscript accordingly to include these details. We have added the following discussion to the revised version of the manuscript at page 7-8.

It is commonly assumed that proteins sharing similar structures also exhibit similar functions. Several measurements have been developed to assess protein structure similarity, each offering unique insights. Root Mean Square Deviation (RMSD)²⁴ quantifies the average spatial variance between corresponding atoms or components within superimposed proteins, providing a fundamental measure of structural deviation. For our analysis, RMSD calculations were performed by superimposing corresponding atomic coordinates using a least-squares fitting procedure implemented in R software, focusing on backbone C α atoms to assess the overall fold without the influence of side-chain orientations. While RMSD is widely used, it heavily relies on direct spatial overlap and is sensitive to outlier regions. This sensitivity often penalizes flexible regions or domain movements, such as hinge motions or flexible loops, potentially obscuring meaningful similarities in overall protein fold when local variations are present. The TM-score (Template Modeling score)²⁵ evaluates similarity by considering both residue-level alignment and overall topology, offering a nuanced assessment of structural resemblance. Unlike RMSD, TM-Score is less sensitive to domain-level movements but has its own limitations. Specifically, when comparing proteins with highly variable structures, TM-Score tends to favor the alignment of larger structural elements, potentially leading to lower scores for proteins with significant domain rearrangements despite sharing similar overall folds. Additionally, TM-Score may not adequately account for functional sites that involve small but critical local structural differences. TM-Vec²⁶, a recent advancement, employs deep learning techniques trained on diverse protein structures to enhance accuracy and efficiency in similarity assessment. TM-Vec maps protein structures into a vector space, allowing comparisons based on the distances between their vector representations. While highly accurate in detecting remote homology and structural similarities, TM-Vec's reliance on training data introduces inherent biases. These biases are particularly evident when evaluating proteins with rare or novel folds that are underrepresented in the training set. Furthermore, as a black-box model, TM-Vec offers limited interpretability regarding the specific structural features contributing to the similarity scores, which can be a limitation when detailed structural insights are required. On the alignment front, we utilized GR-Align²⁷ with its default parameters, employing the graphlet degree similarity (GDS) metric to capture topological similarities between protein structures. The GDS metric compares the distributions of small-connected subgraphs (graphlets) within the protein structures. GR-Align is robust in identifying proteins with similar topological arrangements, but its sensitivity to minor structural variations can lead to higher false positives when comparing proteins with subtle differences in their tertiary structures. Additionally, GR-Align does not incorporate sequence information or account for conformational flexibility,

which may limit its ability to discern functionally relevant structural variations, particularly those involving dynamic regions of proteins. Finally, the Hausdorff distance²⁸ provides a measure of dissimilarity between sets of points, offering further insight into structural comparisons. In our study, the Yau-Hausdorff distance was calculated by comparing point sets representing the protein structures, specifically using distances from the structural centroids to the C α atoms. This method captures overall geometric differences between structures by measuring the maximal deviation between point sets in a bidirectional manner. However, it may not fully account for local structural variations or conformational changes, such as those occurring at active sites or ligand-binding regions, which are crucial for functional similarity. Moreover, the method assumes that global geometric similarity correlates with functional similarity, which may not always hold true, especially for proteins whose function is dictated by specific local conformations.

We have also included additional details about computation time and accuracy on page 20.

The computation times and accuracy metrics for RMSD, TM-Score, GR-align, and Yau-Hausdorff, as presented in Table 1 and Fig. 7, were sourced from Table 2 in the study by Tian et al.²⁸. For this analysis, the results for CPE, SPE, and TM-Vec were generated on the same computational setup used in their study (a system with a 2.4 GHz processor and 8 GB of RAM).

3) The reported computational efficiency of the proposed method is notable. Nevertheless, a discussion on the scalability of the method would be beneficial. Specifically, information on how computational time scales with the number of proteins or the complexity of the dataset would provide a clearer perspective on the practical applicability of the method in larger and more complex scenarios.

ANSWER: We appreciate the reviewer's suggestion. To address this, we performed an additional analysis and added the results at page 16 on Figure 14.

To evaluate the scalability of the CPE method, we performed an additional analysis by randomly selecting protein subsets from the ASTRAL95 dataset, ranging in size from 1,000 to 30,000 proteins (at intervals of 5,000). For each subset, we computed the pairwise distances between proteins using both the CPE and TM-Vec methods, while also recording the processing time per amino acid. This metric represents the total computation time divided by the cumulative number of amino acids across all analyzed protein domains. As shown in Fig. 14, both methods demonstrate a linear increase in computation time per amino acid as the dataset size expands. However, the CPE method exhibits a gentler slope compared to TM-Vec, indicating superior scalability. These findings underscore the efficiency of the CPE method, particularly for handling large, complex datasets, making it highly suitable for high-throughput computational studies.

Fig. 14 | Scalability of CPE and TM-Vec. Processing time per amino acid for subsets from the ASTRAL95 dataset, ranging in size from 1,000 to 30,000 proteins (at intervals of 5,000).

4) The results indicate that the energy profile-based model effectively reconstructs the phylogenetic tree. To strengthen these findings, the inclusion of statistical validation measures, such as confidence intervals or statistical significance tests, would be valuable. This addition would provide more robust support to the full SARS-CoV-2 or for the accuracy of the phylogenetic reconstructions.

ANSWER: We thank the reviewer for the suggestion and do agree. In response we have implemented a bootstrap analysis with $B = 100$ replicates to assess the robustness of the phylogenetic tree reconstructions using the CPE, SPE, TM-Vec, and MSA methods. Additionally, we calculated confidence intervals to statistically validate the accuracy of our results. The bootstrap analysis and confidence intervals for the main branches that distinguish the three species are now presented in the Fig. 10 within the manuscript, providing stronger evidence for the reliability of the energy profile-based model in reconstructing phylogenetic trees. We have also added the following discussion to the Section 3.5 (Page 11) to better cover this suggestion. The complete bootstrap results for all branches can be found in Supplementary Figures S2-S5. We hope these revisions address the reviewer suggestion and strengthen our findings.

Additionally, we performed a bootstrap analysis with 100 replicates to assess the robustness of phylogenetic tree reconstructions using CPE, SPE, TM-Vec, and MSA methods. Confidence intervals were calculated to statistically validate the accuracy of these results. The bootstrap values and confidence intervals for the main branches distinguishing the three species are provided in Fig. 10A-B and 10D,

reinforcing the reliability of the energy profile-based model in reconstructing the phylogenetic tree. Detailed bootstrap results for all branches are available in Supplementary Figures S2-S5.

Fig. 10 | Clustering analysis of spike glycoprotein structures from SARS-CoV, SARS-CoV-2, and MERS-CoV. The dendrograms depict the clustering of spike glycoprotein structures from the three viruses: SARS-CoV, SARS-CoV-2, and MERS-CoV. The clustering is based on pairwise distances calculated from different methods: **A)** protein sequence, **B)** CPE, **C)** TM-Vec, **D)** SPE, **E)** RMSD, and **F)** TM-Score. The leaves of each tree are color-coded to indicate the originating virus for each spike glycoprotein structure. **G)** Displays the ARI values for each method, **H)** shows the running time associated with each method scaled to 12 hours, with an inset zooming in on the region indicated by the dashed circle. The entire circle represents 80 seconds, and **I)** presents the average distance between the three virus groups as calculated by each method.

5) The manuscript reports a discrepancy between the text and Table 1 regarding the accuracy and computation time of the 1-NN classifier based on the distance between profiles' energy (CPE). The text states an accuracy of 97% and a computation time of approximately 3 minutes, whereas Table 1 shows an accuracy of 100% and a computation time of 1 second. The authors should reconcile this discrepancy to ensure consistency and accuracy in the reporting of results.

ANSWER: We would like to thank the reviewer for the rigor. We confirm that the correct accuracy for the 1-NN classifier based on the distance between profiles' energy (CPE) is indeed **100%**, and the computation time is **1 second**, as correctly indicated in Table 1. The previously mentioned accuracy of 97% and computation time of approximately 3 minutes in the text was an error and we apologize for that. We have now updated the text (Page 8) to reflect the correct values as presented in Table 1.

As shown in Table 1 and Fig. 7, both CPE and TM-Vec achieved 100% accuracy in distinguishing between the two protein families. However, the CPE method significantly reduced computation time, completing the task in just one second compared to TM-Vec's 67 seconds.

Table 1 | The accuracy and computation time for 1-NN classifier based on GR-Align, RMSD, TM-score, Yau-Hausdorff distance, TM-Vec, and the distance between profiles of energy SPE and CPE as a measure of protein dissimilarity.

Method	Accuracy	Time
GR-Align	62.3%	2 min
RMSD	59.2%	1 h
TM-Score	61.5%	9 h 20 min
YH (10 Rotation)	70.8%	10 min
YH (2500 Rotation)	81.5%	4h 10 min
TM-Vec	100%	67 sec
CPE	100%	1 sec
SPE	99%	187 sec

6) The authors demonstrate that the CPE method not only matches TM-Vec in accuracy but also exhibits superior computational efficiency and interpretability. The reliance of CPE on energy profiles and pairwise amino acid interactions provides a clearer understanding of protein similarities compared to the black-box nature of deep learning techniques used in TM-Vec. Further commentary on how the CPE approach achieves this enhanced interpretability would be valuable. Specifically, elucidating how the reliance on energy profiles and amino acid interactions facilitates a more straightforward understanding of protein relationships compared to TM-Vec's deep learning model would be beneficial.

ANSWER: We would like to thank the reviewer for highlighting this and do agree with the suggestion. We have now added the following discussion into the Discussion section of the manuscript (Page 16-17).

The CPE method leverages energy profiles derived from pairwise amino acid interactions, where each dimension of the energy vector corresponds to specific amino acid pairs. This structured, physically grounded representation makes the data more intuitive and interpretable because the calculated energy values directly reflect biologically meaningful aspects of protein interactions, such as stability, folding, and molecular dynamics. This clarity allows researchers to trace protein similarities back to the underlying energy landscapes, which are well-established in protein science. In contrast, TM-Vec utilizes deep learning embeddings that, although highly effective, function as a "black box." While TM-Vec can identify remote homologies and structural similarities, the embeddings it generates are abstract and difficult to deconstruct

in a biologically meaningful way. This limits the ability to draw direct connections between the model's outputs and the physical or functional properties of proteins.

The CPE method, by focusing on energy profiles, offers distinct advantages in terms of interpretability. Energy profiles are inherently linked to protein folding, stability, and interaction networks, which are fundamental to biological function. For example, an increase in energy in specific pairwise interactions might suggest destabilizing mutations or conformational shifts that affect protein function. This explicit connection between energy and structural features allows for more transparent insights into how variations in energy impact the overall behavior and evolutionary relationships of proteins. By directly correlating these energy states with functional classifications such as folds, superfamilies, or evolutionary relationships, CPE provides clearer, actionable insights for researchers.

Additionally, because energy profiles can be tied to specific biophysical principles, CPE offers a mechanistic understanding of protein relationships that is often lacking in machine learning models like TM-Vec. In fields such as drug discovery or protein engineering, where understanding the precise molecular interactions is crucial, CPE provides a significant advantage in generating interpretable and actionable data. In conclusion, CPE's reliance on energy profiles provides not only a more interpretable but also a biophysically grounded model of protein similarity. This contrasts with TM-Vec's deep learning approach, which, while powerful, offers less transparency and explainability. CPE's approach is particularly valuable in contexts where understanding the biological and structural principles behind protein behavior is critical, such as in evolutionary studies, disease-related mutation analysis, and drug development.

7) The methods section in general would benefit from a more comprehensive description to facilitate a clearer understanding of the methodology and enable others to replicate the study more effectively.

ANSWER: We acknowledge the importance of providing a more detailed and comprehensive Methods section to ensure clarity and reproducibility of our study. We have now revised the Method section and have incorporated a schematic flowchart (Fig. 1) in the revised manuscript (Page 17-19).

Dataset Preparation

A curated dataset of non-redundant protein chains was generated using PISCES⁴⁷ from the Protein Data Bank (PDB). The dataset was selected based on the following criteria:

- **Pairwise sequence identity:** Less than 50% to ensure non-redundancy.
- **Resolution:** Higher than 1.6 Å to guarantee structural accuracy.
- **R-factor:** Below 0.25 to ensure reliable crystallographic data.
- **Protein length:** Between 40 and 1,000 residues to include proteins of varying sizes while excluding excessively short or long chains.

- **Overlap:** Proteins overlapping with the test sets from this manuscript were removed from the training set.

These filtered proteins were utilized to train and calculate the knowledge-based potential function as follows.

Pairwise Distance-Dependent Knowledge-Based Potential

Knowledge-based potentials are derived from databases of known protein structures and are essential for estimating the energies of pairwise interactions. These potentials can be based on various factors, including distance dependencies, dihedral angles, and accessible surface areas⁸. In this study, we employed a distance-dependent potential function where atomic contacts were identified using the tessellation method^{9, 48, 49} as follows:

Contact Identification:

5. **Representation:** All amino acids in each protein chain were represented by their heavy atoms (excluding hydrogen atoms).
6. **Delaunay Tessellation:** A Delaunay tessellation of the resulting point set was computed using Qhull⁵⁰, identifying neighboring atoms based on spatial proximity.
7. **Defining Contacts:** Two atoms were considered to be in contact if they are connected by an edge in the Delaunay triangulation. This implies that they are not shielded from each other by other atoms, ensuring direct interaction without obstruction.
8. **Distance Shells:** The distances between contacting atoms were divided into 30 discrete shells, starting at 0.75 Å with each shell having a width of 0.5 Å. This binning allows for the extraction of distance-dependent interaction potentials.

Atom Types: A total of 167 atom types were considered by treating non-hydrogen atoms as distinct based on their specific amino acid residues.

Energy Calculation: The potential energy between two atoms i and j at distance d was calculated using the following equation⁷:

$$\Delta E^{ij}(d) = RT \left[\ln (1 + M_{ij}\sigma) - \ln (1 + M_{ij}\sigma \left(\frac{f_{ij}(d)}{f_{xx}(d)} \right)) \right] \quad (1)$$

where RT is constant and equal to 0.582 kcal/mole. M_{ij} is the number of observations for atomic pair i and j , $f_{ij}(d)$ is the relative frequency of occurrence for i and j in distance class d , $f_{xx}(d)$ is the relative

frequency of occurrence for all atomic pairs in distance shell d , and σ is the weight given to each observation. As discussed by Sippl⁷, it was assumed that $\sigma = 0.02$.

The potential energy associated with the interaction of residues A and B denoted by $\Delta E(A, B)$ is estimated by summing the pairwise potentials between the atoms of each of these residues as follows:

$$\Delta E(A, B) = \sum_{i \in A, j \in B} \Delta E^{ij}(d) \quad (2)$$

where the sum is over all atom pairs in contact, identified via the Delaunay triangulation method.

Structural Profile of Energy (SPE): Given the 20 standard amino acids, there are 210 unique amino acid-amino acid interaction types. For each protein structure, a 210-dimensional vector was created to represent the distance-dependent energy interactions between residues. Each dimension corresponds to the energy interaction between a specific pair of amino acid types. This vector is referred to as the **Structural Profile of Energy (SPE)**.

Pairwise Energy Content from Amino Acid Composition

While the knowledge-based potential function relies on having the three-dimensional structure of a protein, many protein structures remain undetermined experimentally. To address this, we developed a method to estimate pairwise energy content based solely on amino acid composition.

Energy Estimation: For each protein S in the training set:

- e_i^S denotes the energy of interactions between all residues of type i and all other amino acids in protein S .
- The estimated energy \widehat{e}_i^S is calculated using:

$$\widehat{e}_i^S = N_i^S \sum_{j=1}^{20} P_{ij} n_j^S \quad (3)$$

where, N_i^S represents the frequency of amino acid type i in the structure S , and $n_j^S = \frac{N_j^S}{L}$, is calculated as the ratio of N_j^S to the total number of amino acids in S , denoted by L , and P is the energy predictor matrix, delineating the dependence of amino acid i 's energy on the j th element within the amino acid composition.

Parameter Optimization: The parameters of each row of matrix P were optimized by minimizing the following objective function for each amino acid type

$$Z_i = \sum_S (e_i^S - \widehat{e}_i^S)^2 \quad (4)$$

By setting the partial derivatives $\frac{\partial Z_i}{\partial P_{ij}} = 0$ for all P_{ij} , a system of linear equations was obtained and solved using the Symbolic Math Toolbox in MATLAB.

Compositional Profile of Energy (CPE): For each pair of amino acid types i and j , the energy E_{ij} was estimated based on amino acid sequence composition:

$$E_{ij} = n_i P_{ij} n_j \quad (5)$$

where P is the energy predictor matrix estimated using equation 4. This results in a 210-dimensional vector representing energy interactions between amino acid types based on composition alone. This vector is termed the **Compositional Profile of Energy (CPE)** and is normalized according to protein length.

Fig. 1 | Development of knowledge-based potential function and profile of energy. A) Construction of the knowledge-based potential function. B) Estimation of the predictor matrix P . C) Construction of the structural profile of energy (SPE) based on protein structure. D) Construction of the compositional profile of energy (CPE) based on protein sequence.

At the beginning of the results section on page 5, we provided an explanation of Fig. 1 as follows:

Knowledge-based potentials are derived from databases of known protein structures. Various potential functions, such as distance-dependent, dihedral angles, and accessible surface energies leverage information from known protein structures to estimate energies of pairwise interactions^{7, 8}. In this study, a knowledge-based potential function was developed using a curated dataset of non-redundant protein chains from the Protein Data Bank (PDB), selected for high structural accuracy and diversity as detailed in the Method section. Pairwise distance-dependent potentials were calculated based on atomic interactions, identified through Delaunay tessellation, with energies derived from the frequency of atomic contacts at various distance intervals. The energy between atom pairs was computed following equation (1) in the Method section (Fig. 1A). Furthermore, an energy predictor matrix was created to estimate the pairwise energy content solely from amino acid composition (Fig. 1B). Given the 20 amino acids in proteins, equation

(2) was applied to represent each protein structure using 210 distinct pairwise interaction types (Fig. 1C), leading to the generation of the 210-dimensional Structural Profile of Energy (SPE). Additionally, equation (5) was employed to compute the Compositional Profile of Energy (CPE) based on protein sequences (Fig. 1D). For each pair of proteins, the Manhattan distance between the profiles of energies is considered a measure of dissimilarity between them.

8) The GITHUB repository contains all the described datasets and code; however, it would benefit from additional documentation. Specifically, further explanation on how to execute and run the scripts would be valuable. Additionally, guidance on how to interpret the outputs of these scripts is necessary for users to fully understand and utilize the provided tools. The repository and the manuscript should include a comprehensive table listing all the datasets used in the manuscript. This table should detail the exact names of the datasets and provide direct pointers to the files containing them. This improvement would greatly enhance the usability and transparency of the repository.

ANSWER: We agree that improving the documentation will enhance the usability of our repository. In response, we have updated the repository by adding two scripts, `CPE.R` and `SPE.R`, along with detailed instructions on how to execute and run them, as well as guidance on interpreting the outputs. Additionally, we have included a comprehensive table in both the repository and the manuscript (Page 20), listing all datasets used in the study with their names and direct links or pointers to the relevant files.

The following is a list of datasets used in this study, along with their locations in the repository:

1. Training Set for Knowledge-Based Potential: (Train_Energy/list_with_chainID_rm_Olaps.txt).
2. Protein Domains in ASTRAL40 (Data/csv/astral-scopedom-seqres-gd-sel-gs-bib-40-2.08.fa) and ASTRAL95 (Data/csv/astral-scopedom-seqres-gd-sel-gs-bib-95-2.08.fa).
3. The list of Bacteriocin Proteins family (Data/csv/Bacteriocin.csv).
4. PDBIDs of Ferritin Superfamily (Data/csv/Ferritin_Like_seq.csv) (SCOP ID: a.25.1).
5. The list of C-terminal and Homing endonucleases (Data/csv/CT_Ho_cathID.csv) (CATH Code: 1.10.8.10 and 3.10.28.10).
6. The list of protein domains of five superfamilies (Data/csv/fiveSF.csv) winged helix (SCOP ID: a.4.5), PH domain-like (SCOP ID: a.55.1), NTF-like (SCOP ID: d.17.4), Ubiquitin-like (SCOP ID: d.15.1), and Immunoglobulins (SCOP ID: b.1.1).
7. Covid19 spike proteins data set (Data/covidPDB/).
8. The list of Drug-Targets (Data/csv/41467_2019_9186_MOESM4_ESM.xlsx) including 65 antihypertensive drugs and their protein targets IDs.
9. The list of 21 mammalian hemoglobin's proteins in Globin family (Data/Globin/Globin.csv).
10. Large-Scale SARS-CoV-2 Proteome Analysis across 28 families. The protein models can be found in the Large_Scale_SARS2 folder (Data/Large_Scale_SARS2) according to the sars_proteom column (Data/Large_Scale_SARS2/sars_proteom.csv).

Comments from the Reviewers:

Reviewer 1:

The authors have addressed most of my comments and concerns. The manuscript has been significantly improved compared to the previous version. There is one additional issue that the author may want to address. There are no details about how the authors perform the protein class prediction and how the different evaluation metrics are calculated throughout the different tasks. They only mentioned that they used kNN (k=1) to make predictions. Are these predictions performed using a leave-one-out procedure or a K-fold cross-validation procedure? Are these predictions formulated as binary or multi-class predictions? It is confusing why the class accuracy in Tables S3, S4, and S5 are exactly the same for different classes of proteins.

ANSWER: We much appreciate the reviewer's additional feedback and are pleased to hear that the manuscript is now significantly improved.

To clarify, the predictions were conducted using a k-Nearest Neighbors (k-NN) classifier with k=1, evaluated through a Leave-One-Out Cross-Validation (LOOCV) approach. In this method, each sample in the dataset serves as a test point while the remaining samples form the training set. This exhaustive evaluation ensures a thorough performance analysis by systematically testing the classifier on every individual sample in the dataset.

For the classification process, we utilized the Manhattan distance to measure similarity between the test sample and all other samples, identifying the nearest neighbor (top match, k=1) and assigning the corresponding class label. This was carried out as a multi-class prediction task, where each protein was classified into one of the predefined classes. To improve clarity, we have now included a detailed description of this procedure in the Methods section of the revised manuscript.

Concerning the accuracy values in Tables S3, S4, and S5 (now Supplementary Tables 5, 6, 7 in the revised version), we acknowledge the oversight. The total accuracy was mistakenly reported in the 'Accuracy' column for each row instead of the class-specific accuracies. This has been corrected, and the tables now accurately reflect the class-specific accuracies for each protein class. Additionally, the total accuracy as average has now been provided in the final row of the tables to improve clarity.

Supplementary Table 5. The results of 1-NN classification on SARS Proteome using CPE.

SARS Proteom	Accuracy	Precision	Recall	F1
E_protein	99,9546	100	96,55	98
N_C-terminaldomain	100	100	100	100
N_N-terminaldomain	99,9773	99,82	100	100
NSP1_protein	99,9773	100	95,83	98
NSP10_protein	100	100	100	100
NSP12_protein	100	100	100	100
NSP13_protein	100	100	100	100
NSP14_protein	100	100	100	100
NSP15_protein	100	100	100	100
NSP16_protein	99,9773	99,61	100	100
NSP2_protein	100	100	100	100

NSP3_cd21525_SUD_C_SARS-CoV_Nsp3	99,9773	99,2	100	100
NSP3_cd21557_Macro_X_Nsp3-like	99,9546	99,19	99,19	99
NSP3_cd21717_TM_Y_SARS-CoV-like_Nsp3_C	99,9773	100	99,19	100
NSP3_cd21732_betaCoV_PLPro	99,9773	99,18	100	100
NSP3_cd21822_SARS-CoV-like_Nsp3_NAB	99,9319	98,33	99,16	99
NSP3_cl00019_Macro_SF	99,9773	100	97,62	99
NSP3_cl13138_SUD-M	99,8638	99,17	95,97	98
NSP3_cl13772_DUF3655	100	100	100	100
NSP5_protein	99,9773	100	99,59	100
NSP7_protein	100	100	100	100
NSP8_protein	99,9773	99,5	100	100
NSP9_protein	99,9092	98,21	99,4	99
orf3a_protein	100	100	100	100
orf7a_protein	100	100	100	100
orf8_protein	100	100	100	100
orf9b_protein	99,9546	91,3	100	95
S_protein	100	100	100	100
Average	99,9773	99,4111	99,375	99,4643

Supplementary Table 6. The results of 1-NN classification on SARS Proteome using SPE.

SARS Proteom	Accuracy	Precision	Recall	F1
E_protein	99,9546	100	96,55	98
N_C-terminaldomain	99,9773	100	99,82	100
N_N-terminaldomain	100	100	100	100
NSP1_protein	99,9773	100	95,83	98
NSP10_protein	99,9773	99,55	100	100
NSP12_protein	100	100	100	100
NSP13_protein	100	100	100	100
NSP14_protein	100	100	100	100
NSP15_protein	100	100	100	100
NSP16_protein	100	100	100	100
NSP2_protein	100	100	100	100
NSP3_cd21525_SUD_C_SARS-CoV_Nsp3	99,9092	96,88	100	98
NSP3_cd21557_Macro_X_Nsp3-like	100	100	100	100
NSP3_cd21717_TM_Y_SARS-CoV-like_Nsp3_C	100	100	100	100
NSP3_cd21732_betaCoV_PLPro	100	100	100	100
NSP3_cd21822_SARS-CoV-like_Nsp3_NAB	99,9546	98,35	100	99
NSP3_cl00019_Macro_SF	99,7957	97,14	80,95	88

NSP3_cl13138_SUD-M	99,9773	100	99,19	100
NSP3_cl13772_DUF3655	100	100	100	100
NSP5_protein	100	100	100	100
NSP7_protein	100	100	100	100
NSP8_protein	100	100	100	100
NSP9_protein	99,9546	98,81	100	99
orf3a_protein	100	100	100	100
orf7a_protein	99,9546	92,86	100	96
orf8_protein	99,9773	95	100	97
orf9b_protein	100	100	100	100
S_protein	100	100	100	100
Average	99,9789	99,2354	99,0121	99,0357

Supplementary Table 7. The results of 1-NN classification on SARS Proteome using TM-Vec.

SARS Proteom	Accuracy	Precision	Recall	F1
E_protein	99,9773	98,31	100	99
N_C-terminaldomain	99,9773	100	99,82	100
N_N-terminaldomain	100	100	100	100
NSP1_protein	99,9773	100	95,83	98
NSP10_protein	99,9773	100	99,55	100
NSP12_protein	100	100	100	100
NSP13_protein	100	100	100	100
NSP14_protein	100	100	100	100
NSP15_protein	100	100	100	100
NSP16_protein	100	100	100	100
NSP2_protein	100	100	100	100
NSP3_cd21525_SUD_C_SARS-CoV_Nsp3	99,8184	94,62	99,19	97
NSP3_cd21557_Macro_X_Nsp3-like	100	100	100	100
NSP3_cd21717_TM_Y_SARS-CoV-like_Nsp3_C	99,9773	100	99,19	100
NSP3_cd21732_betaCoV_PLPro	100	100	100	100
NSP3_cd21822_SARS-CoV-like_Nsp3_NAB	99,9092	98,32	98,32	98
NSP3_cl00019_Macro_SF	99,9773	100	97,62	99
NSP3_cl13138_SUD-M	99,8865	99,17	96,77	98
NSP3_cl13772_DUF3655	100	100	100	100
NSP5_protein	100	100	100	100
NSP7_protein	100	100	100	100
NSP8_protein	100	100	100	100

NSP9_protein	99,9319	98,22	100	99
orf3a_protein	99,9773	100	96,67	98
orf7a_protein	99,9546	100	92,31	96
orf8_protein	99,9773	95	100	97
orf9b_protein	100	100	100	100
S_protein	100	100	100	100
Average	99,9757	99,4157	99,1168	99,25

Reviewer 3:

The authors have thoroughly addressed all of my revisions, and I am happy to recommend the article for publication, pending further edits by the editorial team.

ANSWER: We sincerely thank the reviewer for the time and effort throughout the review process and we are delighted to hear that the revisions have addressed the concerns.